# Spatial, Temporal, and Meteorological Impact of the 26 February 2023 Dust Storm: Increase in Particulate Matter Concentrations Across New Mexico and West Texas

Mary. C. Robinson[1] Kaitlin Schueth[2] and Karin Ardon-Dryer [1]

1 Department of Geosciences, Texas Tech University, Lubbock TX, USA 79409.

2 NOAA/National Weather Service, 2579 S Loop 289, Lubbock, Texas 79423 USA

*Correspondence to*: Karin Ardon-Dryer (karin.ardon-dryer@ttu.edu)

**Abstract.** The Southwestern portions of the United States experience dust events frequently due to the arid and semi-arid environments and close proximity to multiple deserts. On 26 February 2023, a dust event was initiated in New Mexico due to strong winds aloft mixing down to the surface. The dust intensified as it moved eastward into West Texas and developed into a dust storm (visibility < 1 km) for multiple locations. This study examined the meteorological characteristics of this dust storm using 28 meteorological stations and examined the impacts on $PM_{2.5}$ and/or $PM_{10}$ (particulate matter with an aerodynamic diameter < 2.5 and 10 µm) concentrations using 19 air quality stations. The dust event lasted up to ~16 hours, dust storm conditions lasted from five minutes up to 120 minutes. The highest wind speed and wind gust recorded during the dust were 27 m s$^{-1}$ and 37 m s$^{-1}$ respectively. This dust had a strong impact on the air quality in the area, as very high PM values were recorded across the region, and nine of the PM stations exceeded the EPA daily threshold. The maximum hourly $PM_{2.5}$ and $PM_{10}$ concentrations recorded were 518 µg m$^{-3}$ and 9,983 µg m$^{-3}$ respectively. These concentrations at the peak of the dust were an order of magnitude higher than the minimum hourly $PM_{2.5}$ and $PM_{10}$ concentrations recorded on the dust day. The highest hourly $PM_{10}$-$PM_{2.5}$ concentration recorded was $760 \pm 1000$ µg m$^{-3}$, while the lowest hourly $PM_{2.5}$/$PM_{10}$ concentration measured was $0.05 \pm 0.01$.

## 1 Introduction

Dust events are a meteorological phenomenon that occurs when dust particles are suspended in the atmosphere by strong winds and reduce visibility. Visibility during dust events ranges between 1 to 10 km, while dust storms are classified when visibility drops below 1 km (WMO, 2019). Dust events are prominent in arid and semi-arid environments, but can influence other types of environments (Middleton, 2019). The strength of the dust event is dependent on multiple factors. Strong winds are very important for the initiation of dust events and/or storms, which are generally caused by a synoptic or convective meteorological disturbance (Kelley and Ardon-Dryer, 2021; Robinson and Ardon-Dryer et al., 2024; Sandhu et al., 2024). Drought conditions (Arcusa et al., 2020) and vegetation cover (Stout, 2001) are also important factors that contribute to dust initiation.

There are multiple hazards associated with dust events. Lower visibility increases the chances of traffic and aviation accidents (Li et al., 2018; Al Kheder and Al Kandari, 2020; Tong et al., 2023). The blowing dust particles can cause abrasions and

damage to crops (Middleton, 2019) and health complications for livestock (Mu et al., 2013). High particulate matter (PM) concentrations can result in poor air quality (Achakulwisut et al., 2017; Ardon-Dryer and Kelley, 2022; Ardon-Dryer et al.,

2023a), leading to different health and wellbeing impacts. Exposure to dust particles during dust events can cause significant health problems such as respiratory issues (Toure et al., 2019; Herrera-Molina et al., 2023), cardiovascular issues (Goudarzi et al., 2017), stroke (Schweitzer et al., 2018), toxaemia of pregnancy (Bogan et al., 2021), Valley Fever (Tong et al., 2022; Gorris et al., 2023), and even lead to death (Pérez et al., 2008; Malig and Ostro, 2009). Therefore, the United States Environmental Protection Agency (EPA) and the World Health Organization (WHO) have set standards for $PM_{10}$ and $PM_{2.5}$

(particulate matter with an aerodynamic diameter below 10 and 2.5 µm, respectively) to determine poor air quality conditions. The EPA $PM_{2.5}$ and $PM_{10}$ daily standards are 35 µg m$^{-3}$ and 150 µg m$^{-3}$, respectively (EPA, 2023), while the WHO updated daily thresholds are 15 µg m$^{-3}$ and 45 µg m$^{-3}$ for $PM_{2.5}$ and $PM_{10}$, respectively (WHO, 2023).

Dust events and storms occur across the United States (Tong et al., 2023), mainly across the southwestern portions, due to its

drier and warmer conditions with low soil moisture from desert regions (Achakulwisut et al., 2017). Among the states, the most susceptible to dust events are Arizona (Nickling and Brazel, 1984; Lei et al., 2016; White et al., 2023), southern California (Bach et al., 1996; Evan, 2019; Huang et al., 2022), Utah (Hahnenberger and Nicoll, 2012; Hennen et al., 2022) and states across the Great Plains, mainly the Southern Great Plains area including New Mexico and Texas (Kandakji et al., 2020; Hennen et al., 2022; Ardon-Dryer et al., 2023b; Robinson and Ardon-Dryer, 2024). The multiple dust sources in the region, mainly

cropland, contribute to the high number of dust events (Lee et al., 2012). In a recent study, Robinson and Ardon-Dryer (2024) found an average of 22 dust events annually (between 2000 to 2021) across four dust-prone regions in West Texas. Most of the dust events in the region occur in the spring to early summer months, mainly due to synoptic disturbances, while a smaller percentage of dust events are formed by convective disturbances, and rarely are dust events formed by the combinations of these two disturbances (Robinson and Ardon-Dryer, 2024).


The air quality across West Texas and New Mexico is good overall (Zanobetti and Schwartz, 2009; Kelley et al. 2020). Anthropogenic pollution such as industrial facilities and transportation emissions, which can lead to Ozone, can be found mainly in the two large urban cities of El Paso, Texas, and Albuquerque, New Mexico (Gaffney et al., 1997; Chen et al., 2012; Kavouras et al., 2015; Craig et al., 2020; Karle et al., 2020; Van Pelt et al 2020; Huang et al., 2023). The entire area is impacted

by dust events and dust storms which lead to an increase in PM and degradation of the air quality (Tong et al., 2012; Stout, 2015; Herrera-Molina et al., 2021; Kelley and Ardon-Dryer, 2021; Ardon-Dryer et al., 2023b; Albuquerque-Bernalillo County, 2024). In Sunland Park, New Mexico, Li et al. (2005) found during dust events that $PM_{2.5}$ and $PM_{10}$ hourly concentrations were 170 µg m$^{-3}$ and 2346 µg m$^{-3}$, respectively, while daily averages were $12 \pm 8$ µg m$^{-3}$ and $69 \pm 72$ µg m$^{-3}$, respectively.

Kelley et al. (2020) analysed $PM_{2.5}$ concentrations in Lubbock, Texas over 17 years (2001 – 2018) and found that the majority of the hourly $PM_{2.5}$ concentrations were lower than 10 µg m$^{-3}$ (80%), but there were several days with high PM including 15

April 2003 and 15 December 2003 that had $PM_{2.5}$ hourly values of 433 and 486 µg m$^{-3}$, respectively. Rivera Rivera et al. (2009) also examined the impact of these two dust storms in El Paso and found on 15 April 2003 hourly $PM_{10}$ concentrations of 4724 µg m$^{-3}$ with a daily $PM_{10}$ concentration of 375 µg m$^{-3}$, while the hourly $PM_{10}$ concentrations on 15 December 2003, was >1200 µg m$^{-3}$. Daily $PM_{10}$ concentrations on 15 December 2003, for another site in Texas, was >160 µg m$^{-3}$ (Tong et al., 2012). Yin et al. (2005) examined hourly $PM_{2.5}$ and $PM_{10}$ measurements from different stations across New Mexico and Texas during the same dust storm (15 December 2003). They found hourly $PM_{10}$ concentrations in New Mexico >700 µg m$^{-3}$, while $PM_{2.5}$ hourly concentrations ranged from 12 up to 36 µg m$^{-3}$ (Yin et al., 2005). Both of these dust storms were caused by synoptic disturbances. In Lubbock Texas, it was found that $PM_{2.5}$ daily concentrations during synoptic dust events had slightly higher $PM_{2.5}$ average concentrations compared to convective dust events. Ardon-Dryer and Kelley (2022) also found that synoptic dust events had higher $PM_{2.5}$ and $PM_{10}$ daily concentrations compared to convective dust events, but short-term observation (based on 10 minutes) showed that convective have much higher PM concentrations. The impact of $PM_{2.5}/PM_{10}$ and $PM_{10}$-$PM_{2.5}$ values during, dust events in the region, were examined but only by a handful of studies. In New Mexico, $PM_{2.5}/PM_{10}$ values ranged from 0.05 up to 0.58, and the $PM_{2.5}/PM_{10}$ ratio was extremely low (0.079 up to 0.093) during dust events (Li et al., 2005). Measurements of daily $PM_{2.5}$ and $PM_{10}$ using multiple Interagency Monitoring of Protected Visual Environments (IMPROVE) stations in New Mexico and Texas also found a significant drop in the $PM_{2.5}/PM_{10}$ ratio during dust events, with daily means that ranged from 0.22 to 0.24 during dust events (Tong et al., 2012).

The dust storm of 26 February 2023 was one the strongest dust storms that occurred in this region over the last two decades. This study aimed to understand the meteorological conditions that initiated this dust storm and those measured during it using multiple meteorological stations across New Mexico and Texas, capturing its Spatial and Temporal changes. The impact this dust storm had on air quality over the two states was of interest to understand if and how significant its impact on PM concentrations in the region was and to evaluate its similarity to previous dust events in this region.

## 2. Methods

### 2.1 Automatic Surface Observation Station (ASOS)

Automatic surface observation systems (ASOS) are meteorological stations located at most airports across the United States that provide meteorological measurements released to the public via Meteorological Aerodrome Reports (METARs). The meteorological measurements include 5-minute to 1-hour measurements of temperature, dew point, relative humidity, wind speed, wind direction, wind gust, pressure, visibility, precipitation, and a Present Weather code. Some stations are continuously monitored by a contracted weather observer while others are mostly automatic (Ardon-Dryer et al., 2023b). The purpose of the weather observer is to back up any instrumentation outages and augment any weather information the ASOS station cannot automatically record. The Present Weather code (entered automatically by the station or manually by the on-duty observer) is

an important aspect of the METAR as it provides information on the current weather, such as thunderstorms, fog, hail, and dust events. The classification of the dust event in this study was based on the combination of present weather codes such as

BLDU (blowing dust), VCBLDU (vicinity blowing dust), DU (widespread dust), DS (dust storm), and HZ (haze), with the reduction of horizontal visibility (< 10 km) and increase of wind speed (> 6 m s$^{-1}$) but without precipitation, similar to the method used in Ardon-Dryer et al. (2023b) and Robinson and Ardon-Dryer (2024). The different present weather codes for dust are defined by the World Meteorological Organization (WMO) and the Federal Aviation Administration (FAA). BLDU represents a case when the dust is present in the atmosphere and visibility drops below 11 km, DU indicates that dust is present

and gives distant objects a tan or gray tinge, DS represents when dust drops the visibility to 1 km or less, and VCBLDU refers to that the dust is present within 8 to 16 km away from the station. Each of these codes can only be entered manually by a weather observer (WHO, 2019; FAA, 2021). It should be noted that 16.1 km is the maximum visibility that should be reported by the ASOS (ASOS User's Guide, 1998). Many studies have used the present weather codes to identify dust events in this region (Kandakji et al., 2020; Herrera-Molina et al., 2021; Kelley and Ardon-Dryer, 2021; Robinson and Ardon-Dryer, 2024).

METAR data from 28 ASOS stations across West Texas and New Mexico were downloaded from the Iowa University Mesonet (Iowa Mesonet, 2023) for February 2023. Table S1 provides information on each of the ASOS stations utilized in this study, while Fig. 1A shows their location. Additional ASOS stations (13 in total) from the region could not be used, since they either had missing measurements or they did not have dust conditions on February 26. It should be noted that there are some limitations to the use of ASOS stations, as there were only four stations with full-time weather observed, while the remaining

were semi-/fully automated. In a recent study (Robinson and Ardon-Dryer, 2024), it was found that there could be mistakes in dust identifications. To make sure such will not happen, this study follows the guidance provided by Robinson and Ardon-Dryer (2024) to remove such cases. Another related issue is the fact that the automated stations can stop operating due to outages and in some cases, cannot be backed up, which has happened to some of the stations in this study limiting the ability to use the data. Regardless of these limitations, the usage of the ASOS with the spatial and temporal coverage allowed

examination into the development and movement of the dust event.

## 2.2 Particulate Matter

Hourly concentrations for PM$_{10}$ and PM$_{2.5}$ from 6 stations across West Texas were taken from the Texas Commission on Environmental Quality (TCEQ, 2023), while hourly PM$_{10}$ and PM$_{2.5}$ concentrations for 13 stations across New Mexico were taken from the New Mexico Environment Department (New Mexico Environmental Department, 2023) or provided by Mr.

Patrick Hudson, a Senior Environment Health scientist for the City of Albuquerque. All of the PM sensors are Federal Equivalent Methods (FEMs). Each FEM instrument had a different resolution depending on the operated unit, some units ranged from 0.1 up to 10,000 µg m$^{-3}$ (T640, 2024), or -15 up to 10,000 µg m$^{-3}$ (BAM 1022, 2024), others had an upper limit of 5,000 µg m$^{-3}$ (R & P Model 2025; EPA, 2024), see Table 2 for information on instrument used at each location. The PM data included hourly measurements for February 2023. Calculations of the daily average were made for each day based on

hourly measurements from midnight to 23:00 local time (LT). Table S2 outlines each of the PM sensors used in this study, while Fig. 1B shows the geographical spread of PM sensors. Six stations only measured $PM_{2.5}$ (5ZS, 6Q, C320, C1025, C1014, and C37) concentrations, and four only measured $PM_{10}$ (6ZK, 6ZL, 6ZM, and 6WM) concentrations, while the remaining nine stations had both $PM_{2.5}$ and $PM_{10}$ (6CM, Del Norte HS, Foothills, Jefferson, North Valley, San Jose, South Valley, C49, and C41). The majority of the stations across West Texas contain only $PM_{2.5}$ measurements while other locations including El

Paso, Texas, and Albuquerque, New Mexico contain both $PM_{2.5}$ and $PM_{10}$. Calculations of $PM_{10-2.5}$ and $PM_{2.5}/PM_{10}$ were performed for stations that contained both $PM_{2.5}$ and $PM_{10}$. Almost all PM sensors had meteorological measurements such as hourly ambient temperature, wind direction, and wind speed. If these variables were not available, meteorological measurements from the nearest ASOS or PM station were used. For example, station 6Q in Las Cruces, New Mexico did not have wind measurements and therefore wind measurements from the closest station (6WM in West Mesa, which is 10.6 km away) were used to supplement the missing data. Additionally, station C320 in Amarillo, Texas did not have wind

measurements and therefore wind measurements were taken from the Amarillo (AMA) ASOS station (11.8 km away). Lastly, the San Jose and North Valley (in Albuquerque, New Mexico) stations did not have wind measurements and therefore wind measurements from the South Valley station (5.7 and 5.2 km away, respectively) were used. It should be noted (as shown in Fig. 1B) that there is a wide spatial gap between the PM sensors, as these are the only active sensors in the area. Also, most of

the PM sensors in Texas (except those in El Paso) only provide $PM_{2.5}$ meaning the impact of $PM_{10}$ in West Texas will not be provided in this study.

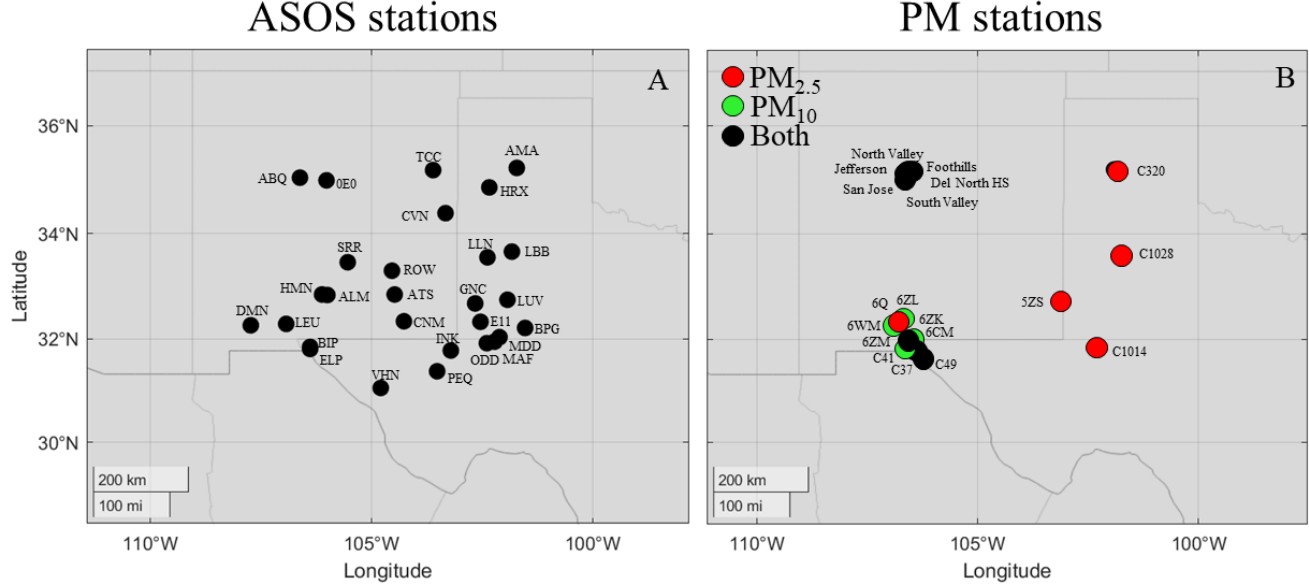

**Figure 1. Distribution of ASOS (A) and PM (B) stations spread across New Mexico and West Texas used in this study.**

## 2.3 Meteorology Overview Maps

The synoptic maps were made using the North American Rapid Refresh version 3 (RAPv3) with a horizontal grid spacing of 13 km and 51 vertical levels (Benjamin et al., 2016). The RAPv3 was selected to illustrate the meteorology due to its one-hour assimilation frequency and ability to provide one of the best forecasts in the rapidly changing atmosphere. Only the initialization hours were hours used in this study. Each synoptic map was made using the Metpy python package (May et al., 2023), with several meteorological variables layered. The following variables were chosen to analyze the meteorology; geopotential heights (mid-level and surface), wind speed and direction (mid-level and surface), temperatures (mid-level), and dewpoint temperatures (surface).

The Geostationary Operational Environmental Satellite (GOES) imagery over eastern North America (GOES-East, also known as GOES-16) encompasses the research area and provides 5-minute updates. The satellite data was pulled from Amazon Web Services and plotted with the GOES-2-Go python package (Blaylock, 2022). The Dust RGB derived satellite product was utilized to highlight the progression of the dust throughout the dust event. The Dust RGB, which consists of band differencing and IR thermal channel, allows dust to be observed through satellite imagery during both the day and night. The density of the dust particles was inferred by the range of the magenta/pink colour. This method is commonly used to detect and identify dust events (Fuell et al., 2016; Ardon-Dryer et al., 2023b).

Nexrad WSR-88D radars were also used to visualize thunderstorms and mesoscale boundaries. The data from various radars across the Southern Plains (Lubbock, Midland/Odessa, Amarillo, Clovis, San Angelo, Dyess Air Force Base, and Frederick) was retrieved via Amazon Web Services to plot a mosaic radar image using a Pyart python package (Helmus and Collis, 2016).

## 3. Results

### 3.1 The Meteorological Conditions Resulted in the Formation of the Dust Storm

During the morning and afternoon hours on 26 February 2023, a robust and slightly negatively tilted 500 mb closed low ejected eastward from Southern California across the Four Corners region of the United States (Fig. 2A). The right exit region (Fig. 2B) of the nearly 51-62 m s$^{-1}$ (100-120 knot) 500 mb jet streak, associated with the upper low, entered the Chihuahuan Desert region of Mexico, Texas, and New Mexico around early to mid-afternoon . A stacked 700 mb trough axis brought a 31-36 m s$^{-1}$ (60-70 knot) jet axis over the area at the same time (data not shown). In conjunction with the approaching upper low, intense surface cyclogenesis developed along the leeward side of the Rockies before sliding eastward across the Oklahoma Panhandle and then pushing farther northeast into Kansas by midnight. Warm air advection from southerly surface winds led to sufficient daytime heating and mixing of the strong winds aloft to the surface. In addition, the south winds advected low level moisture

into the Southern Plains with a weak dryline present and near 40-to-50-degree dewpoints east of the boundary (Fig. 2C). Instability in the atmosphere combined with this boundary led to thunderstorm development across the far southern Texas Panhandle through the afternoon hours (starting around 17:00 LT; 23:00 UTC). As the upper level closed low continued to swing eastward through the evening, so did the corresponding north-south extending Pacific cold front. The front eventually caught up with the dryline just east of Lubbock where additional storms initiated along the colliding boundaries as shown in

the radar reflectivity (Fig. 3A), along with both boundaries (dryline and Pacific cold front) at 17:00 Central time for Texas (23:00 UTC). These thunderstorms created very strong wind gusts up to 51 m s$^{-1}$ across West Texas (shown in Fig. S1).

    The south-southwest winds began to increase through the morning hours and into the afternoon, with several severe wind gusts (> 26 m s$^{-1}$) reported across eastern New Mexico and West Texas. Multiple National Weather Service (NWS) offices across

the Southern High Plains and Southern Great Plains highlighted the wind potential through products such as High Wind Warnings and Wind Advisories. As the south-southwest surface winds increased, dust particles began to be lofted and started to cause a reduction in visibility. When the Pacific front began to move eastward across New Mexico, winds began to shift out of the west and continued to exhibit strong to severe wind speeds. Additional dust particles were suspended along the quick moving boundary. Both radar reflectivity and satellite imagery (Fig. 3) reveal the evolution and intensity of the dust along this

front during the afternoon hours across West Texas. One of the biggest factors that made this event an anomaly was the already advected dust across a large area (~4×10$^5$ km$^2$) prior to the frontal passage due to the strong to severe winds from the south-southwest during the morning hours, as shown in Fig. 3B. Some of the locations mentioned throughout this study (e.g., Lubbock, Texas) might experience both synoptic and convective disturbances, which may assist in the duration of high wind gusts along with the severity of dust in the region. The fact that some of the locations had both synoptic and convective

disturbances (also known as combinations) is a rare aspect of this region, as only a handful of the dust events were caused by such conditions, For Lubbock Texas, ~15% of the past DS (2000-2021) were caused by a combination of disturbances (Robinson and Ardon-Dryer, 2024).

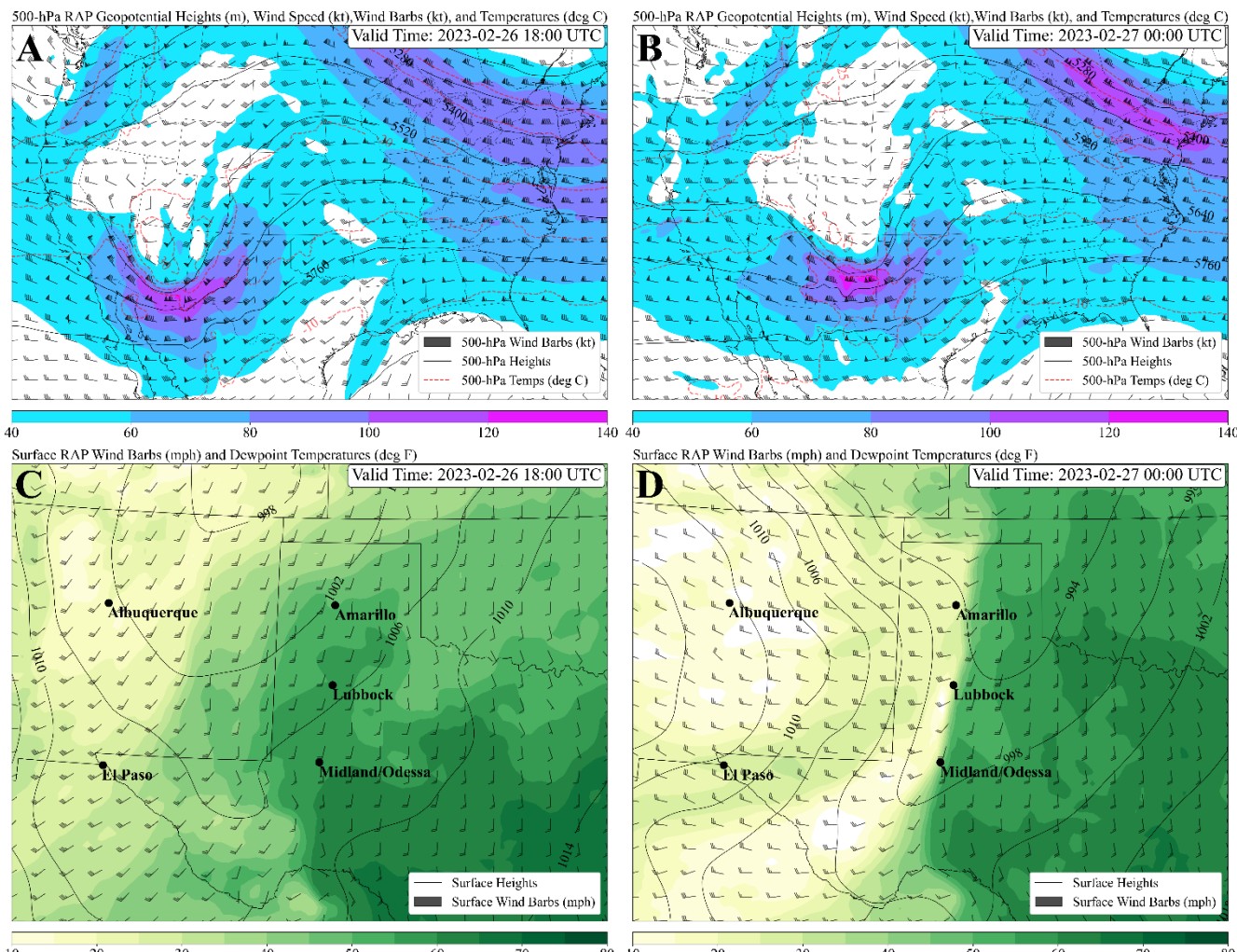

**Figure 2. 500 mb geopotential heights (m), wind speed (kt, shaded), wind barbs (kt), and temperature (°C) for 26 February 2023 at 18:00 UTC, 12:00 central time, when the dust event started (A) and 27 at 00:00 UTC, 18:00 central time, when the dust event intensified and turned into a dust storm across west Texas (B) and surface wind barbs (mph) and dew point temperature (°C, shaded) for February 26 at 18:00 UTC, 12:00 central time (C) and 27 at 00:00 UTC, 18:00 central time (D).**

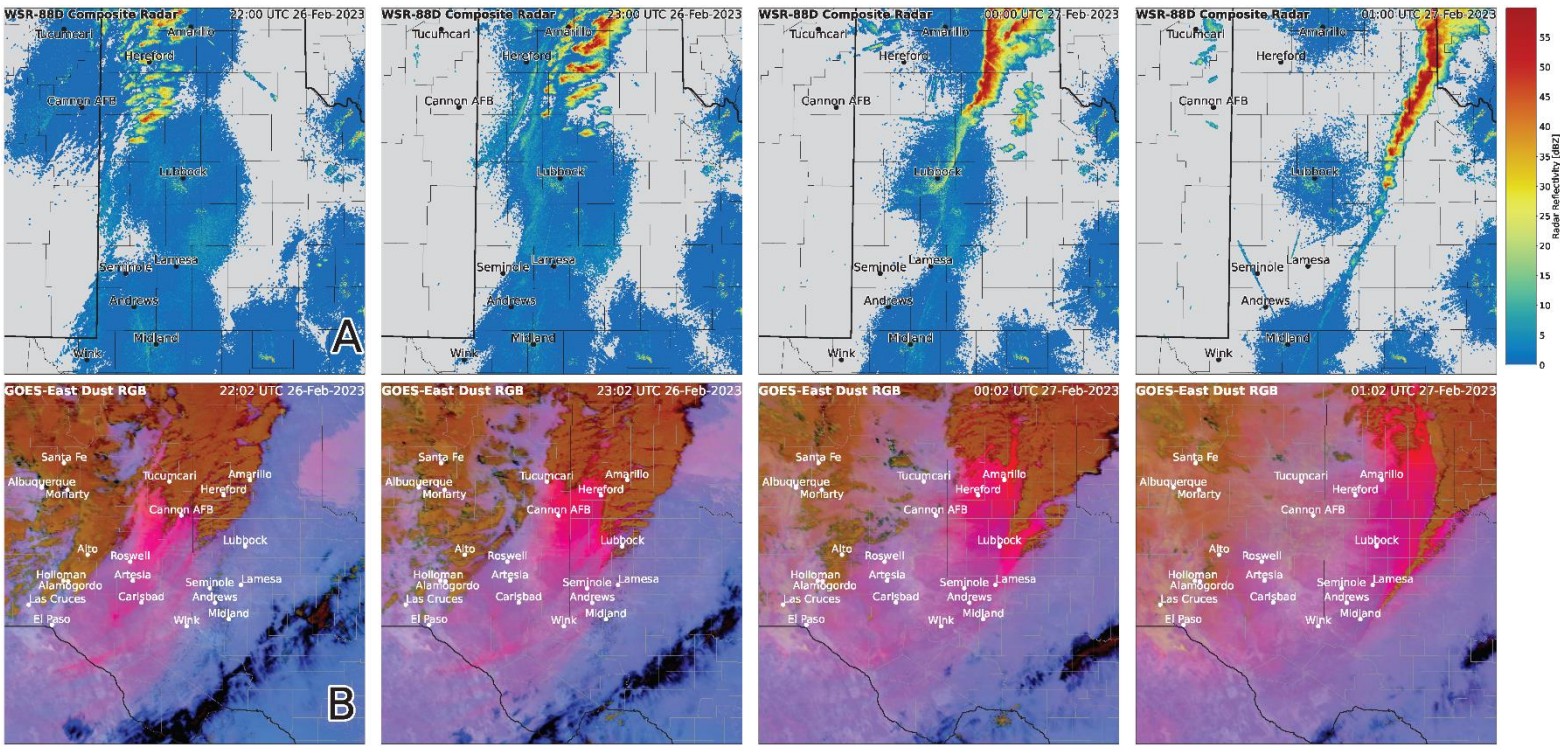

**Figure 3. Radar imagery (A) and GOES-East Dust RGB (B) of the dust storm from 26-27 February at four different times (16:00, 17:00, 18:00 and 19:00 central local time; 22:00, 23:00, 00:00 01:00 UTC, respectively).**

## 3.2 Meteorological Observations During the Dust Storm

Observations of wind speed, wind gust, and visibility were collected from each of the 28 active ASOS units. ASOS units with on-duty observers (such as LBB for Lubbock, ELP for El Paso, and ABQ for Albuquerque) had present weather codes that represent dust (e.g., BLDU, VCBLDU, DS and SS, Sand Storm), while the remaining automated ASOS units had a present
weather code of haze (HZ). Some of the stations reported a dust weather code before the visibility was reduced (<10 km), but since this work follows the WMO (2019) dust events definition, these times were not included. The first observation of dust was when the visibility was reduced below 10 km, and the end of the dust event was when visibility first exceeded 10 km. Observations of visibility showed that the dust event started in New Mexico during the morning hours and was first observed in Texas around noon local time. The first report of dust across New Mexico in the present weather code was in Deming
(DMN) at 10:00 LT, with HZ reported and visibility dropped below 10 km to 6.4 km, making it officially a dust event based on the World Meteorological Organization (WMO, 2019). At 10:21 LT in Albuquerque (ABQ) visibility was reduced to < 10 km at 10:21 LT. In Texas, the first observations of dust were in El Paso (ELP) at 11:10 LT when the visibility dropped to 8.0 km. The dust moved northeast and reached the West Texas region about an hour later. The last station that had observation of dust was Midland-Odessa (MDD). Despite the strong winds that started in the morning hours, the first reduction of visibility
only happened at 18:35 LT. All three Midland-Odessa stations experienced the dust at a later time when the Pacific cold front and dryline moved across the area. The last report of dust was on February 27 at 04:55 LT at the LLN station in Levelland, Texas. The ASOS stations LBB and LUV, which are near LLN, also had reports of dust until after 03:00 LT. The presence of dust at some of the stations (based on duration with visibility <10 km) varies. On average the dust across all stations lasted for 5:30± 3:30 hours since some stations experienced dust for short duration, while others for long durations. The station with the
shortest duration was MDD where the dust lasted only 80 minutes. For the ASOS in New Mexico, the average duration of dust was 4:07 hours, ranging from 1:40 hours (0E0) up to 6:27 hours (HNM). In Texas, on average the reported dust was 6:42 hours, with the shorter duration in MDD, and the longest duration 15:20 hours in LUV.

The duration of the dust events on February 26 varied between stations over many hours from 1:20 hours up to 15:20 hours.
Although overall this dust event lasted longer than previous dust events reported in the area (Doggett IV et al., 2002; Kelley and Ardon-Dryer, 2021). Shorter dust events were also reported in Arizona (Nickling and Brazel, 1984; Raman et al., 2014; Eagar et al., 2017) and in other locations around the world such as China (Wang et al., 2005; Guan et al., 2015) and Turkmenistan (Orlovsky et al., 2005). Yet, some of the durations reported by other studies were similar to those found in this study. For example, Novlan et al. (2007) found for El Paso that most dust events lasted on average 3–4 hours, and the longest
lasted for 24 hours. Robinson and Ardon-Dryer (2024) examined hundreds of dust events across West Texas and found that most of the dust events lasted less than 5 hours, which is around the average duration found in this study. Although many of

the dust events were less than 1 hour the longest dust event reported was even longer than the one reported here. Similar duration times have been reported in other locations, for example, in Mongolia, dust events last on average 2-6 hours (Natsagdorj et al., 2003). Yet, long lasting dust events (two days and more) have been reported in other locations around the world such as in the Middle East (Birinci et al., 2023) and Europe (Sorribas et al., 2017).

The wind speed measured at the beginning of the dust event (first visibility observation <10 km) ranged from 12 m s$^{-1}$ (MAF) up to 26 m s$^{-1}$ (SRR at Alto, New Mexico) with wind gusts measured from 16 m s$^{-1}$ (MAF) up to 33.4 m s$^{-1}$ (SRR at Alto, New Mexico). The highest wind speed and wind gust (27 m s$^{-1}$ and 37 m s$^{-1}$, respectively) recorded during the dust event was in New Mexico (TCC at Tucumcari). Calculation of the average wind speed and wind gust for the duration of the dust event was performed for each station. The average wind speed during the duration of the dust events across all the ASOS stations was 17 $\pm$ 2 m s$^{-1}$, these values vary from 12 $\pm$ 3 m s$^{-1}$ up to 22 $\pm$ 2 m s$^{-1}$. The average wind gust reported across all the ASOS stations at the time of the dust was 23 $\pm$ 3 m s$^{-1}$, these values vary from 17 $\pm$ 3 m s$^{-1}$ up to 29 $\pm$ 2 m s$^{-1}$. Information from each ASOS station including duration, maximum wind speed, and wind gust at the beginning and during the dust event can be found in Table S3. These wind speeds and wind gusts measured during the dust events were 3.2 times higher than the average wind speed and wind gust recorded in the month of February 2023 (shown in Table S3). The difference was much stronger for the strongest recorded wind speed and wind gust, up to 5.9 times and 8.3 times (respectively) compared to the month of February. These big differences indicate how strong this dust event was. But looking at the overall meteorological conditions during this month, it seems that there were additional dust events during that month (e.g., February 9 and 22), but were not as strong as the one reported here (data not shown). Perhaps if these dust times had been removed from the monthly analysis the difference between the meteorological conditions would have been stronger.

Most of the wind speeds at the beginning of the dust event were above the wind speed reported by Stout and Arimoto (2010) as a threshold for dust to be suspended. The wind speeds during this event were in the range of wind speeds reported in Hagen and Woodruff (1973) for dust events that occurred in the Great Plains in the 1950s. All ASOS locations examined showed wind speed values at the beginning of the dust event that were higher than the wind speed reported by Zobeck and Van Pelt (2006) during the March 2003 dust storms in the region. Similar maximum wind speed values were measured in the area during the 15 December 2003, dust storm (Lee et al., 2009). However, much higher wind speeds were measured in the July 2014 dust storm in Phoenix, Arizona (Eagar et al., 2017), perhaps since the one in Arizona was mainly convective.

The reduction of visibility during the dust event varied from station to station, the average visibility for the duration of the dust event was 5.4 $\pm$ 1.5 km, with these values varying from 3.2 $\pm$ 2.3 up to 9.2 $\pm$ 5.4 km. The average of the lowest visibility reported during the dust event was 1.5 $\pm$ 1.0 km. Although all the ASOS units in the region experienced dust event conditions, not all experienced dust storm conditions (visibility < 1 km), as shown in Table S3. A total of 11 ASOS stations (six in New Mexico and five in Texas) reported visibility below 1 km and therefore experienced dust storm conditions. The remaining

stations had a reduction in visibility but not below 1 km (shown in Table S3). The visibility values of stations that experienced dust storm conditions ranged from 0.8 km down to 0 km, as shown in Fig. 4. The duration of the dust storm conditions ranged from 5 minutes (INK) up to 2:00 hours (LLN). The ASOS station in Lubbock, Texas (LBB) reported 0 km visibility for a continuous 13 minutes, which highlights the severity of this dust storm. It should be noted that several of the West Texas

stations that experienced DS conditions (including LBB, LLN, LUV, and INK) had fog conditions in the morning hours of this dust event day, which could explain the low visibility values in the morning hours (Fig. 4). Almost all stations showed a peak wind speed and wind gust at the time of minimum visibility, as shown in Fig. 4. The western stations (New Mexico) experienced dust storm conditions around noon LT when wind speeds reached their maximum. Meanwhile, the eastern stations (mainly those in West Texas) experienced dust storm conditions in the late afternoon when the front collided with the dryline.

Satellite observations from the GOES-16 Dust RGB (shown in Fig. 3B) highlight the high concentrations of dust particles during these times. Satellite observations from GOES-16 showed that the dust particles from this dust storm made it to Oklahoma, Kansas, and Arkansas (data not shown). Previous studies also found that dust particles from this region can of travel to neighbouring states including Oklahoma (Park et al., 2007; Kandakji et al., 2020) and as far as the northeastern states, even into Canada (Doggett IV et al., 2002; Park et al., 2007) depending on the synoptic setup.


This region of New Mexico and West Texas is prone to dust events (Park et al., 2009; Kandakji et al., 2020; Kelley and Ardon-Dryer, 2021) due to the proximity to the Chihuahuan Desert and many agriculture fields (Rivera Rivera et al., 2009; Lee et al., 2012). Studies found that many of the dust events in this region occur during December – May and more so in the springtime months (Stout, 2001; Novlan et al., 2007; Rivera Rivera et al., 2009). Severe dust storms have been observed in the past in this

region (Lee and Tchakerian, 1995; Lee et al., 2009). Lee et al. (2009) analyzed the 15 December 2003, dust storm that started in New Mexico and moved eastward through West Texas. The dust storm was caused by an upper low-pressure system that brought in a cold front. During this dust storm, wind gusts in Lubbock, Texas were over 28 m s$^{-1}$ and visibility was reduced to 0.4 km. In El Paso, Texas the minimum visibility reported was 2.8 km and wind gusts reached 23 m s$^{-1}$ (Lee et al., 2009). The DS conditions during the event presented in this work were more severe compared to the DS on 15 December 2003, presented

in Lee et al. (2009). The conditions of the current DS were also stronger (higher maximum wind gust and lower visibility values) than those reported in El Paso during the 15 April 2003, dust storm (Park et al., 2009; Rivera Rivera et al., 2009). Strong dust storms with similar meteorological conditions have been observed in other locations across the United States, including Arizona (Raman et al., 2014; Kim et al., 2017) and Utah (Nicoll et al., 2020). In Utah, Nicoll et al. (2020) examined a dust storm that occurred on 14-15 April 2015. An intense intermountain cyclone caused high wind gusts (up to 35 m s$^{-1}$) and

dust storm conditions, with visibility down to 0.4 km. In Arizona, dust storms are common (Lei et al., 2016; Ardon-Dryer et al., 2023b). One of the largest and most famous dust storms recorded in Arizona (near Phoenix and Tucson), occurred on July 5, 2011 (Raman et al., 2014; Vukovic, et al., 2014; Lader et al., 2016). This dust storm developed due to thunderstorms (Lader et al., 2016) and had a peak wind gust of 29 m s$^{-1}$ and visibility of 0 km (Raman et al., 2014; Vukovic, et al., 2014), similar to the conditions presented in this work.

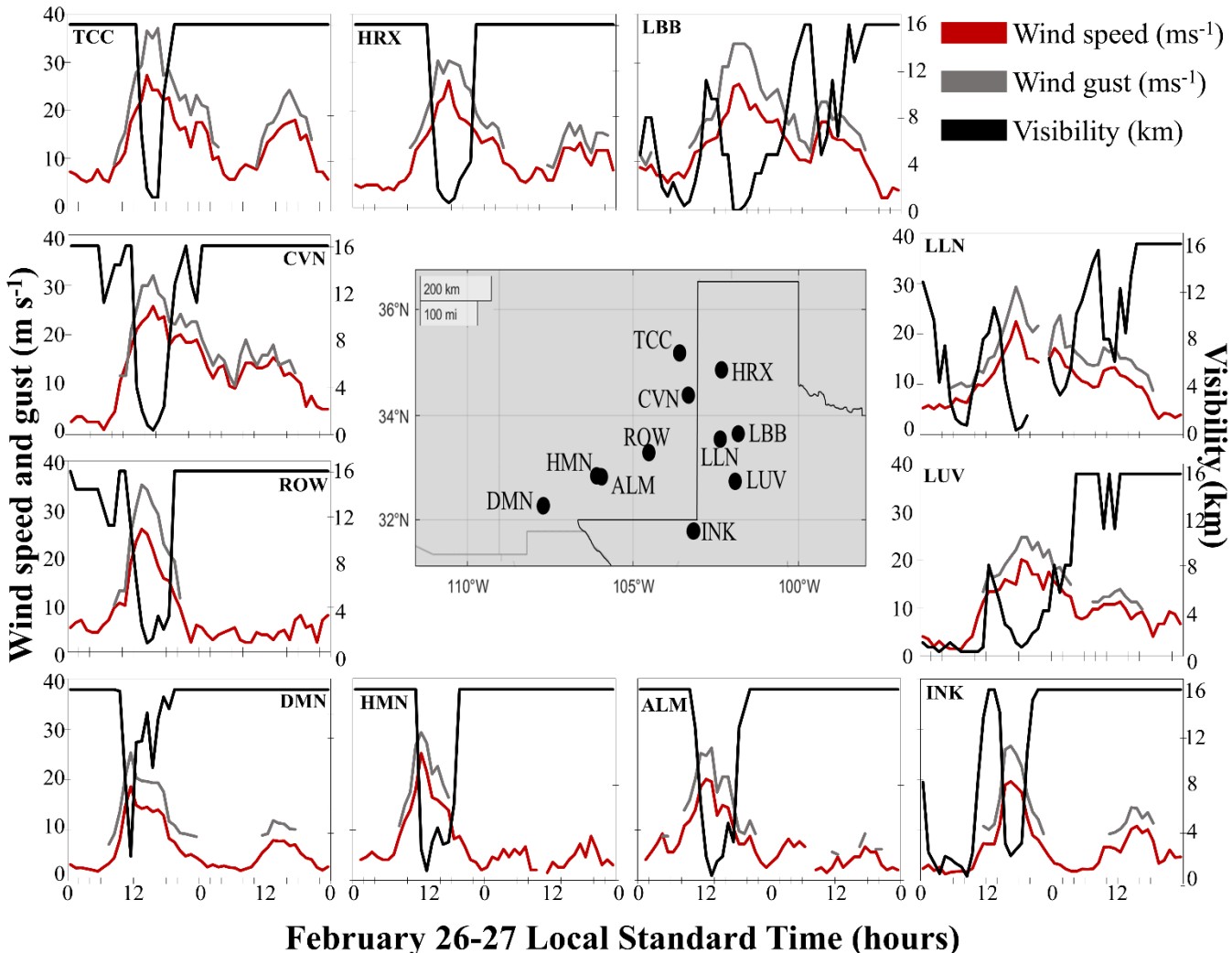

**Figure 4. Observations of wind speed (red) and wind gust (gray) as well as visibility (black) from ASOS stations that experienced dust storm conditions (visibility < 1 km).**

### 3.3 Impact of Dust Storm on PM Concentrations and Air Quality

Nineteen air quality stations were active across the New Mexico and West Texas region during the dust event presented in this study. Each of the PM stations showed an increase in PM values during the dust event, but not all PM stations had a strong impact by the dust event as indicated by the varying hourly PM measurements across the region (Fig. 5). The maximum PM values at the peak of the dust for each station ranged from 14 μg m$^{-3}$ (Foothills station in Albuquerque, New Mexico) up to

518 µg m$^{-3}$ (station C1028 in Lubbock, Texas) for PM$_{2.5}$ and from 198 µg m$^{-3}$ (Foothills station in Albuquerque, New Mexico) up to 9,983 µg m$^{-3}$ (station 6ZM in Desert View, New Mexico) for PM$_{10}$. Details for each station can be found in Table S4. High PM concentrations during dust storms are common in this area (Ardon-Dryer et al., 2022a,b; Kelley and Ardon-Dryer, 2021), along with other locations across the United States (Hahnenberger and Nicoll, 2012; Lei et al., 2016; Achakulwisut et al., 2017) and around the world (Ardon-Dryer and Levin, 2014; Mamouri et al., 2016; Arhami et al., 2017; Milford et al., 2020).

The duration of dust particles in the air was based on the time from the first increase in PM to the decrease in PM values. This duration was similar to the duration of reduction of visibility, mentioned in section 3.2. These durations based on PM values varied, some stations had an increase in PM values for a duration of 2 hours, while others for up to 12 hours. Despite the reduced visibility to 1.6 km during the dust event at the Albuquerque ASOS (ABQ), most of the PM stations in the area witnessed a small increase in PM$_{2.5}$ but a more significant increase in PM$_{10}$ concentrations (as can be seen in Fig. 5 and Table S4). A spatial impact of the dust was also observed in Albuquerque, as stations in the southern part of Albuquerque had higher PM concentrations (with a stronger increase) compared to those located in the northern part of Albuquerque. When calculating the increased ratio of PM, which is indicated by the ratio of PM concentrations at the peak of the dust compared to the PM concentrations right before the dust, results showed an increase in PM across the region, even across Albuquerque. PM$_{2.5}$ concentrations during the dust event were on average 12.8 times higher compared to the time before the dust event (ratios vary from 3.0 up to 36.3), while PM$_{10}$ concentrations during the dust event were on average 216.9 times higher compared to before the dust event (ratios vary from 11 up to 1426). When we examined the same ratio aspect for the lowest PM concentrations recorded on February 26 (shown in Table S4) the differences were much higher, PM$_{2.5}$ and PM$_{10}$ concentrations at the peak of the dust were higher by more than an order of magnitude (on average) than the minimum daily PM concentration recorded on the same day.

Higher hourly PM$_{2.5}$ concentrations for Lubbock, Texas (518 µg m$^{-3}$ at peak of dust) were measured during this dust storm compared to those measured during the past dust storm of 15 December 2003, when PM$_{2.5}$ concentrations reached 486 µg m$^{-3}$ (Lee et al., 2009; Park et al., 2009). The hourly PM$_{2.5}$ concentration measured in Lubbock was higher than any hourly PM$_{2.5}$ concentration recorded over the last 20 years. In El Paso Texas, during the same 15 December 2003 dust events the hourly PM$_{10}$ concentration was >1200 µg m$^{-3}$ (Rivera Rivera et al. 2009), which was higher compared to the hourly PM$_{10}$ concentration measured in El Paso (999 µg m$^{-3}$ by station C41) during this dust storm. However, several New Mexico PM stations measured concentrations >9868 µg m$^{-3}$ (PM stations 6ZM and 6ZL). Most of the maximum hourly PM$_{10}$ concentrations in this study were higher compared to the hourly maximum PM$_{10}$ concentration measured during three previous dust events in 2019 in Lubbock, Texas (Ardon-Dryer and Kelley, 2022). Many of the hourly maximum PM$_{2.5}$ concentrations measured by Ardon-Dryer and Kelley (2022) were higher than the hourly PM$_{2.5}$ concentrations presented in this study. Observations of hourly PM$_{2.5}$ and PM$_{10}$ concentrations in this study were in a similar range to PM concentrations for other dust

storms measured across the United States. For example, Nicoll et al. (2020) examined a dust storm that occurred across the Great Basin region of Utah in April 2015, where $PM_{2.5}$ and $PM_{10}$ hourly concentrations reached 298 µg m$^{-3}$ and 890 µg m$^{-3}$, respectively. Measurements of PM concentrations during various dust storms in Arizona also showed similar values of PM in comparison to the results of this dust event (Raman et al., 2014; Kim et al., 2017). There were some dust events in Arizona with much higher PM hourly concentrations (Eagar et al., 2017; Hyde et al., 2018) compared to this study. For example, Raman et al. (2014) reported high PM values in Phoenix, Arizona during a convective dust event, with hourly maximum $PM_{2.5}$ and $PM_{10}$ concentrations of 907 µg m$^{-3}$ and 1974 µg m$^{-3}$, respectively. These different PM values could be attributed to the differences in regions, along with the cause of the dust event.

Observations of the changes in PM concentrations and wind speed as shown in Fig. 4 indicate a relationship between the two variables. Some studies found a correlation between wind speed and PM concentrations (Karami et al., 2017; Kim et al., 2017), while others could not find a strong relationship between the two (Kelley and Ardon-Dryer, 2021). Calculations of regression (linear and polynomial) were made based on hourly PM concentrations and wind speeds from the PM stations with measurements from February 26 and showed a low linear correlation for most of the stations (Table S4). For stations that measured $PM_{2.5}$ concentrations, $R^2$ values for linear regression were not significant and ranged from 0.01 (North Valley station in Albuquerque, New Mexico) up to 0.47 (station 5ZS in Hobbs, New Mexico, and station C49 in El Paso, Texas). For stations that measure $PM_{10}$ concentrations, $R^2$ values ranged from 0.3 (Jefferson station at Albuquerque, New Mexico) up to 0.6 (station C49 in El Paso, Texas). Only three $PM_{10}$ stations (station C49 in El Paso, Texas, station 6CM in Anthony, New Mexico, and North Valley station in Albuquerque, New Mexico) had high linear correlation values ($R^2 \geq 0.5$). Other regression models were also examined, to potentially find a better regression value between wind speed and PM values. The Polynomial regression (with 2nd-degree polynomial) presented much higher $R^2$ values compared to a linear regression. With $R^2$ values that ranged from 0.37 up to 0.9 for $PM_{2.5}$ and from 0.18 up to 0.9 for $PM_{10}$. 73.3% of the $PM_{2.5}$ stations and 84.6% of the $PM_{10}$ stations had $R^2 \geq 0.5$ (see $R^2$ values in Table S4).

The calculations of PM concentrations during the time of dust were performed for each station. The PM concentrations during the time of dust, which varied per station were on average $70 \pm 50$ µg m$^{-3}$ for $PM_{2.5}$ and $686 \pm 689$ µg m$^{-3}$ for $PM_{10}$. The $PM_{2.5}$ concentrations during the time of dust ranged from $8.4 \pm 4.0$ µg m$^{-3}$ in the Foothills station in Albuquerque, New Mexico up to $154 \pm 135$ µg m$^{-3}$ at station C1028 in Lubbock, Texas. The $PM_{10}$ concentrations during the time of dust ranged from $104 \pm 65$ µg m$^{-3}$ at the Foothills station in Albuquerque, New Mexico up to $2354 \pm 3745$ µg m$^{-3}$ at station 6ZL in Desert View, New Mexico (values for each sensor are shown in Table S4). The $PM_{2.5}$ and $PM_{10}$ concentrations during the time with dust were 2.0 and 3.3 times lower (for $PM_{2.5}$ and $PM_{10}$, respectively) compared to those measured at the peak of the dust. Calculation of daily values for each station (calculated for February 26 from midnight to 23:00 LT for each sensor) showed a wide range of values. $PM_{2.5}$ daily concentrations ranged from $3.8 \pm 2.7$ µg m$^{-3}$ (Foothills station in Albuquerque, New Mexico) up to $69 \pm 121$ µg m$^{-}$

[3] (station C1028 in Lubbock, Texas), while $PM_{10}$ daily concentrations ranged from $28 \pm 42$ μg m[-3] (Foothills station in Albuquerque, New Mexico) up to $748 \pm 2090$ μg m[-3] (station 6ZM in Desert View, New Mexico). On average the PM concentrations during the time of the dust were 3.2 times higher compared to the daily concentrations (for both $PM_{2.5}$ and

$PM_{10}$), the difference ranged from 1.4 to 5.7 for $PM_{2.5}$ and from 2.0 up to 7.6 for $PM_{10}$, as shown in Table S4. The concentrations at the peak of the dust were also higher compared to daily values. The peak of the dust had concentrations 6.1 and 10.5 times higher (average for $PM_{2.5}$ and $PM_{10}$, respectively) compared to the daily average, the difference ranged from 2.4 (for North Valley) up to ~12 (for 6CM) for $PM_{2.5}$ and from 4.0 (for North Valley) up to 21 (for San Jose) for $PM_{10}$, Similar differences of higher PM concentrations at the peak of dust compared to daily values are shown in Hahnenberger and Nicoll (2012) for

Utah dust storms when the daily concentrations were 3.8 and 12 times lower (for both $PM_{2.5}$ and $PM_{10}$, respectively) compared to those measured at the peak of the dust.

To examine the impact of the 26 February 2023 dust event on the overall PM concentrations, daily PM concentrations were calculated, for each PM sensor, for each day during February 2023 (shown in Fig. S2). The daily average for February 26

seems high (for most sensors) compared to the other February day's daily average, it also seems to have much higher SD values compared to manty of the other days. The lowest impact seems to be in the Albuquerque stations, perhaps since the area is also impacted by anthropogenic pollution. The southern part of New Mexico and many of the stations in West Texas seem to have had a bigger impact on this dust event, as daily values for the dust day (February 26) were on average 12 times higher compared to the overall daily $PM_{2.5}$ concentrations and 28 times higher compared to the $PM_{10}$ daily concentrations. These

differences could have been higher, but it seems there were additional pollution events (other dust events, as indicated above) in some of the locations, which increased the daily PM concentration for some days in some of the stations. Observations of daily $PM_{2.5}$ concentrations from the different Albuquerque PM stations show that the dust was not as strong as it was for other locations such as South New Mexico and West Texas. Next, the monthly $PM_{2.5}$ and $PM_{10}$ concentrations for the entire February month were calculated (for each sensor), without February 26 PM concentrations (Table S4). The monthly $PM_{2.5}$ and $PM_{10}$

concentrations were on average 4.0 and 9.3 times lower, for $PM_{2.5}$ and $PM_{10}$ concentrations respectively, compared to the daily concentrations measured on February 26. The monthly PM concentrations were 14 and 27 times lower compared to the $PM_{2.5}$ and $PM_{10}$ concentrations (respectively) measured during the time of the dust, and 26 and 105 times lower (for $PM_{2.5}$ and $PM_{10}$ concentrations, respectively) compared to the PM concentrations at the peak of the dust. These large differences between the concentrations of $PM_{2.5}$ and $PM_{10}$ during the dust to those over the month indicate that while the background PM across the

region might be low (except for Albuquerque) dust events in this region can have a significant impact on both $PM_{2.5}$ and $PM_{10}$ concentrations in the region which will impact on the air quality wellbeing and people's health.

Since EPA and WHO refer to air quality levels based on daily values, the daily concentrations for both $PM_{2.5}$ and $PM_{10}$ on February 26 were examined and compared to the EPA and WHO daily threshold. Only nine PM stations (three $PM_{2.5}$ and six

$PM_{10}$) exceeded the EPA daily thresholds (35 μg m[-3] for $PM_{2.5}$ and 150 μg m[-3] for $PM_{10}$), indicated by the red daily average

values in Fig. 5. Five of them were in southern New Mexico (6ZK, 6ZM, 6ZL, 6CM, and 6WM) and the remaining four were in West Texas (Lubbock, Amarillo, and two in El Paso). The stations where $PM_{10}$ daily values exceeded the EPA daily threshold ranged from $205 \pm 321$ µg m$^{-3}$ (station C49 in El Paso, Texas) up to $748 \pm 2090$ µg m$^{-3}$ (station 6ZM in Desert View, New Mexico). $PM_{2.5}$ daily values for stations that exceeded the EPA daily values ranged from $36 \pm 40$ µg m$^{-3}$ (station C320 at

Amarillo, Texas) up to $69 \pm 121$ µg m$^{-3}$ (station C1028 in Lubbock, Texas). Analysis based on the new WHO thresholds for $PM_{2.5}$ (daily values of 15 µg m$^{-3}$) and $PM_{10}$ (daily values of 45 µg m$^{-3}$), showed that nine of the $PM_{2.5}$ stations (60%) and 11 of the $PM_{10}$ stations (85%), were above the WHO thresholds considered these locations experience a bad air quality level. Many studies use daily average to represent the PM concentration during dust (Tong et al., 2012; Ardon-Dryer and Levin, 2014; Achilleos et al., 2016; Reynolds et al., 2016; Milford et al., 2020), which makes sense when the dust lasts for many

hours or even longer than a day (Krasnov et al., 2016; Sugimoto et al., 2016). However, many of the dust events across the United States last for only a short period of a few hours or less (Claiborn et al., 2000; Kelley et al., 2020; Joshi, 2021; Robinson and Ardon-Dryer, 2024). As shown in this analysis and suggested in Ardon-Dryer et al. (2023a), the daily values underestimate and mask the concentration of both $PM_{2.5}$ and $PM_{10}$ concentrations. It is important to have hourly concentration measurements, as studies during dust events from this region (El Paso and Lubbock) have shown that maximum daily PM concentrations can

lead to significant increases in hospitalizations on the day of dust and the following days. Different health impacts observed included diseases such as respiratory, asthma, mental, stroke, and many others (Herrera-Molina et al., 2021; 2024).

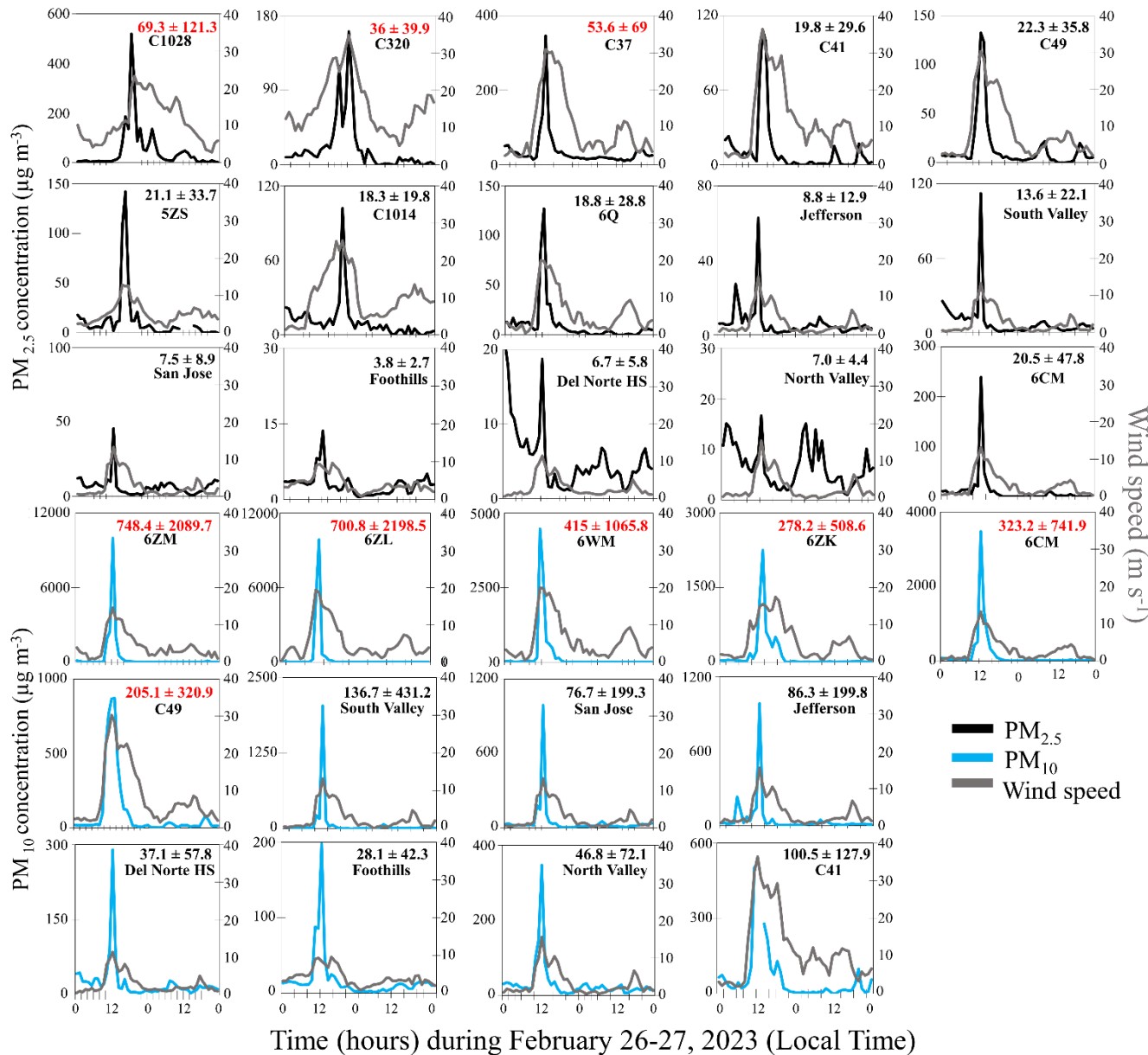

**Figure 5. Changes in PM$_{2.5}$ (black) and PM$_{10}$ (blue) with wind speeds (grey) measured during the dust storm. The name of the station and daily average ± SD values for February 26 are presented in black. Daily average ± SD values for stations that exceeded the EPA daily standards are presented in red.**

Next, the impact of this dust event on the presence of coarse particles (PM$_{10}$-PM$_{2.5}$) and the ratio between PM$_{2.5}$/PM$_{10}$ was examined. Across the study area, there were only nine stations that measured both PM$_{2.5}$ and PM$_{10}$. The majority (six) were in Albuquerque, while the remaining three were around the El Paso area. Calculations of PM$_{2.5}$/PM$_{10}$ and PM$_{10}$-PM$_{2.5}$ were made for each sensor for every hour. High PM$_{10}$-PM$_{2.5}$ values indicate the presence of coarse dust particles in the air. In Albuquerque,

coarse particle concentrations were present in the air for 3-5 hours with concentrations that ranged from 37 $\mu g\ m^{-3}$ up to 2031 $\mu g\ m^{-3}$. The southern part of Albuquerque had a stronger impact by the dust storm as these stations (San Jose, Jefferson, and South Valley) showed higher $PM_{10}$-$PM_{2.5}$ values, with maximum values of 923 $\mu g\ m^{-3}$ up to 2031 $\mu g\ m^{-3}$ at the peak of the dust. This could explain the small increase of $PM_{2.5}$ and $PM_{10}$ found for the northern Albuquerque stations, as it seems the dust was not that strong in that area. In the El Paso area, the $PM_{10}$-$PM_{2.5}$ values during the dust event had a slightly wider range from 53 $\mu g\ m^{-3}$ to 3236 $\mu g\ m^{-3}$. At the peak of the dust, $PM_{10}$-$PM_{2.5}$ values ranged from 871 $\mu g\ m^{-3}$ to 3236 $\mu g\ m^{-3}$ (sensor C41 is missing 2 hours of data during the peak of the dust). The presence of coarse particles in the El Paso area (stations C41, C49, and 6CM) lasted for 8-10 hours, as shown in Fig. 6. Daily $PM_{10}$-$PM_{2.5}$ for February 26 were lower (3.8 up to 7.3 times, average of 5.8) compared to the $PM_{10}$-$PM_{2.5}$ calculated during the time of dust (Table S5). Also, $PM_{10}$-$PM_{2.5}$ at the peak of the dust was 16.6 times higher compared to the daily values. Calculations based on $PM_{10}$-$PM_{2.5}$ for each station for February showed that most stations had a low impact of coarse particles (except for San Jose and 6CM, which had higher monthly $PM_{10}$-$PM_{2.5}$ values, most likely due to the other dust events earlier that month). Both the $PM_{10}$-$PM_{2.5}$ values during the time of dust were higher (3.1 to 19 times, 12 on average) than the $PM_{10}$-$PM_{2.5}$ monthly values. Even the daily $PM_{10}$-$PM_{2.5}$ averages for February 26 were higher (on average 2.2 times) than the monthly values. The hourly $PM_{10}$-$PM_{2.5}$ values from this study were higher, for most stations, compared to values measured in three different dust events in Lubbock Texas, perhaps because this dust event was stronger (Ardon-Dryer and Kelley, 2022). The $PM_{10}$-$PM_{2.5}$ values were higher than those reported in the Rocky Mountains (Reynold et al., 2016) and Utah (Hahnenberger and Nicoll, 2012). Similar ranges of $PM_{10}$-$PM_{2.5}$ values were measured during dust storms in Israel (Krasnov et al., 2016). The daily values were lower compared to those measured in Israel, although the values at the peak of the dust were in the same range (Ardon-Dryer and Levin, 2014). However, the peak $PM_{10}$-$PM_{2.5}$ values were lower compared to the average $PM_{10}$-$PM_{2.5}$ values measured during multiple dust storms in China (Shao and Mao, 2016). Daily $PM_{10}$-$PM_{2.5}$ values in this dust event (for some of the stations) were in a similar range to those measured by Tong et al (2012), who examined multiple dust events in the same area as the one in this study.

Observations based on $PM_{2.5}$/$PM_{10}$ were also performed. The $PM_{2.5}$/$PM_{10}$ ratio is an important indicator used to characterize the underlying atmospheric processes within the local environment, which allows for the identification of the source of the particles (Yu and Wang, 2010). Higher $PM_{2.5}$/$PM_{10}$ ratios (> 0.6) are generally associated with anthropogenic pollution, while lower ratios are associated with dust events (Jugder et al., 2014; Sugimoto et al., 2016; Jaafari et al., 2018; Fan et al., 2021; Ardon-Dryer et al., 2022b). $PM_{2.5}$/$PM_{10}$ values across the nine sensors decreased during the dust event mainly between 11:00 to 18:00 LT (Fig. 6). $PM_{2.5}$/$PM_{10}$ values across the nine stations ranged from 0.03 to 0.13 with an average of $0.07 \pm 0.02$ across all stations and times. These ratios were lower compared to the values reported by Tong et al. (2012), which were 0.22-0.24, for this area combined with multiple dust events. Since Tong et al. (2012) $PM_{2.5}$/$PM_{10}$ values were based on daily values calculations of daily values for each sensor were made (Table S5). The daily $PM_{2.5}$/$PM_{10}$ values were in the same range (and even slightly higher, 0.24 -0.3) as those in Tong et al. (2012). However, observations of these ratios during the time of dust

(which were shorter than the duration of the day, as discussed above) were lower, with the average $PM_{2.5}/PM_{10}$ value of 0.07 (values across all stations ranged from 0.05 to 0.09). These values were similar to those measured by Li et al. (2005) during dust events in the El Paso region. The hourly $PM_{2.5}/PM_{10}$ values at the peak of the dust were lower compared to those measured at the peak in multiple dust storms in Utah (Hahnenberger and Nicoll, 2012; Nicoll et al., 2020). In Washington state, a similar range of daily $PM_{2.5}/PM_{10}$ values was measured (Claiborn et al., 2000). The daily $PM_{2.5}/PM_{10}$ values were in a similar range to those measured during dust events around the world (Alghamdi et al., 2015; Malaguti et al., 2015; Sugimoto et al., 2016; Jaafari et al., 2018).

Observations of the monthly $PM_{2.5}/PM_{10}$ values for February (without the February 26 day, shown in Table S5), ranged from $0.1 \pm 0.08$ (for 6CM) up to $0.43 \pm 0.24$ (for C41). Most of the stations had lower monthly values compared to the daily $PM_{2.5}/PM_{10}$ values, some stations had similar values of ~1. The February 26 daily $PM_{2.5}/PM_{10}$ values were on average 3.6 times lower than the monthly values while the $PM_{2.5}/PM_{10}$ values at the peak of the dust were on average 6.2 times lower. The difference was slightly higher when monthly $PM_{2.5}/PM_{10}$ values were calculated without all the other suspected dust events (as mentioned in section 3.2).

It seems that in some of the locations, the contribution of coarse particles was more crucial than those of fine particles as shown by the low $PM_{2.5}$ and high $PM_{10}$ concentrations, and by the high $PM_{10}$-$PM_{2.5}$ values and low $PM_{2.5}/PM_{10}$ ratios (at least for the stations that had measurements for both $PM_{2.5}$ and $PM_{10}$). However, several of the stations showed higher $PM_{2.5}$ concentrations during the dust events, even 5 times higher (as C1028, in Lubbock). This location and many of the others only contain measurements of $PM_{2.5}$ leading to speculation if the lower contribution for $PM_{2.5}$ would be across the region or just in sites examined (the majority of them were in an urban site). Additional studies are needed during dust events and dust storms across the region to provide measurements for both $PM_{10}$ and $PM_{2.5}$. Additional measurements of particle size distribution are important, as such information will give critical knowledge related to health impact (inhalation of particles into the respiratory system), as well as on radiation and perhaps on cloud formation and precipitation processes.

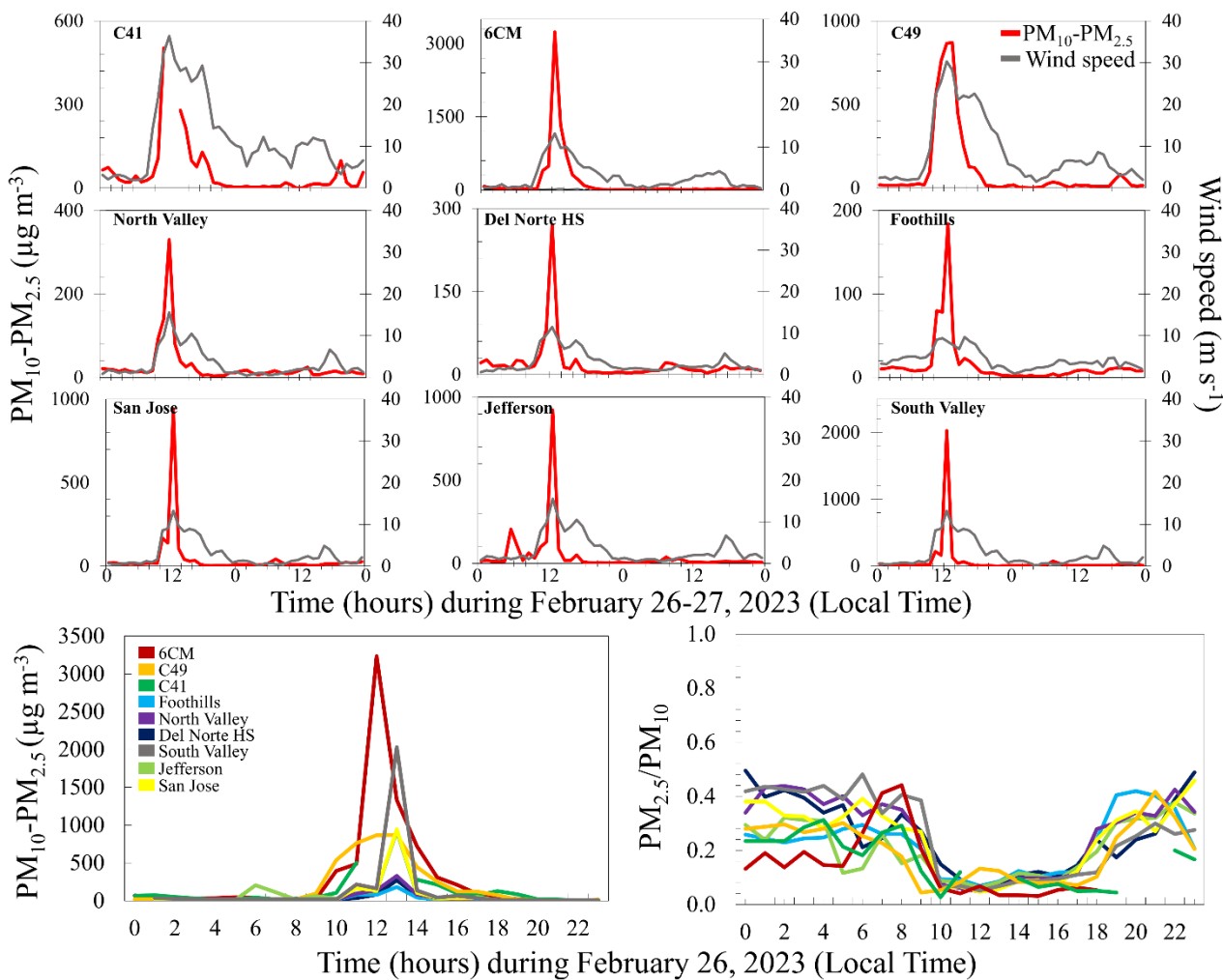

**Figure 6. Measurements of PM$_{10-2.5}$ concentrations (in red) and wind speed (in gray) for each station. PM$_{10}$-PM$_{2.5}$ concentrations and the ratios of PM$_{2.5}$/PM$_{10}$ from stations, each colour represents a different station.**

**4 Conclusion**

On 26 February 2023, an upper-level low-pressure system with a strong jet streak aided in the mixing of strong winds to the surface, which resulted in the formation of a dust storm over portions of New Mexico and West Texas. The dust first initiated in New Mexico during the morning hours and intensified as it moved eastward into West Texas. The average wind speed at the beginning of the dust storm was 15.6 m s$^{-1}$ and during the dust storm wind speeds reached up to 26.2 m s$^{-1}$ with wind gusts up to 37 m s$^{-1}$. Similar wind speeds were measured during different dust storms across the Great Plains, yet lower wind speeds were measured during several dust storms in Arizona. Visibilities ranged from 4 km down to 0 km defining the event as a dust storm (visibility < 1 km). 11 ASOS stations reported dust storm conditions for about 5 to 120 minutes, Lubbock ASOS reported

zero visibility for 13 minutes. This dust storm had a big impact on the air quality in the area. Daily PM concentrations that exceeded the EPA daily threshold ranged from $36 \pm 40\ \mu g\ m^{-3}$ up to $69 \pm 121\ \mu g\ m^{-3}$ for $PM_{2.5}$ and $205 \pm 321\ \mu g\ m^{-3}$ up to $748 \pm 2090\ \mu g\ m^{-3}$ for $PM_{10}$. Nine PM stations exceeded the EPA daily threshold. High hourly $PM_{2.5}$ and $PM_{10}$ concentrations during the dust storm reached a maximum of $518\ \mu g\ m^{-3}$ and $9983\ \mu g\ m^{-3}$ respectively. $PM_{10}$-$PM_{2.5}$ at the time of the dust, based on nine stations ranged from $96 \pm 61\ \mu g\ m^{-3}$ up to $760 \pm 1000 \mu g\ m^{-3}$, which is approximately 6 times higher than the

daily $PM_{10}$-$PM_{2.5}$ values and 12 times higher than monthly $PM_{10}$-$PM_{2.5}$ values. $PM_{2.5}/PM_{10}$ during the dust time, ranged from $0.05 \pm 0.01$ up to $0.09 \pm 0.03$, which were 3.6 times lower than the daily and monthly $PM_{2.5}/PM_{10}$ values. The PM stations in the region, especially in West Texas, are spaced and far apart meaning that higher PM concentrations than those measured could have occurred but not been reported. Dust particles were present in the air for approximately 16 hours impacting millions of citizens across eastern New Mexico and West Texas. In some locations (e.g., Lubbock), this dust storm was the strongest

ever reported, as it had the highest $PM_{2.5}$ concentrations recorded since the station became operational in 2001 and the lowest visibility recorded during a dust storm since 2003. Perhaps the meteorological disturbances that initiated the dust for Lubbock (synoptic with convective) led to these high PM concentrations. Additional studies across the region are needed to understand how meteorological disturbances that initiate dust events might impact the PM concentrations, as such information could be critical for prediction purposes which will help alert the public. Such information could determine whether long-term effects

such as land usage and climate change will affect the frequency and intensity of dust storms in this region.

## Data availability

Automatic surface observation systems (ASOS) are available from the Iowa University Mesonet (Iowa Mesonet, 2023). PM measurements for Texas were retrieved from the Texas Commission on Environmental Quality (TCEQ, 2023), while PM

measurements from New Mexico were downloaded from the New Mexico Environment Department (New Mexico Environmental Department, 2023). PM for Albuquerque was provided form Mr. Patrick Hudson, a Senior Environment Health scientist for the city of Albuquerque's air quality program monitoring section. All measurements are available from the authors upon request.

## Author contributions

KS performed the dust event meteorological overview. MR performed the meteorological data analysis from ASOS and PM analysis. KAD designed the study and coordinated the different aspects of the manuscript. All authors were actively involved in interpreting the results and in discussions on the manuscript.

**Competing interests**

The contact author has declared that none of the authors has any competing interests.

**Acknowledgment**

This research did not receive any specific grant from funding agencies in the public, commercial, or not-for-profit sectors. The authors would like to thank Texas Tech University for the support of Mary Kelley's scholarship and to Mr. Patrick Hudson a Senior Environment Health scientist from the city of Albuquerque, for providing us with the Albuquerque $PM_{2.5}$, $PM_{10}$, and wind speed measurements.

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
