# Peer review of "Spatial, Temporal, and Meteorological Impact of the 26 February 2023 Dust Storm: Increase in Particulate Matter Concentrations Across New Mexico and West Texas"

_EGUsphere, 2024_

## Referee Comment (RC1)

Review of "The Spatial and Temporal Impact of the Dust Storm During February 26, 2023, on Meteorological Conditions and Air Quality Across New Mexico and West Texas" by M.C. Robinson, K. Schueth, and K. Ardon-Dryer

*General Comments:*

This paper examines the meteorological conditions and air quality impacts of a severe dust storm in New Mexico and West Texas in February 2023. Multiple observational datasets are used to characterize the features of the event and associated weather conditions. It is found that the upper-level jet streak, the passage of a cold front, and the formation of thunderstorms along the dryline all contribute to the high wind speeds during the event. The resultant visibility reduction and dramatic increase in PM values highlight the severity of the event. Overall, the study provides a timely and detailed analysis of an extreme dust storm (e.g., the highest $PM_{2.5}$ record at Lubbock, Texas, in the past 20 years), which can potentially advance the current understanding of severe dust storms in the southern U.S. However, I found a few aspects that can be further improved. See my comments below for details.

*Specific comments:*

1. The introduction section can be improved by adding a brief review of dust storms in the southwestern U.S., particularly over New Mexico and western Texas, as background information and by adding a few lines to highlight the motivation and novelty of this study. For instance, what are the key research questions that would be addressed in this study?

2. It would be great to add some discussion about the uncertainties of the datasets used in the study, especially the ground measurements, if possible, and how those may affect the analysis and results.

3. I think Figs. S1-S4 contain information that helps better understand the analysis and should be moved to the main text, given that currently only four figures are in the main text.

4. It would be interesting to add some analysis or discussion about the physical mechanisms and unique aspects of this dust storm, for instance, what caused the strong winds? Lines 131-133 provide some discussion, but it would be interesting to show more if possible.

5. Line 49, airports over certain regions, or the whole U.S.?

6. Lines 57-58, a severe storm with heavy precipitation can also reduce visibility and increase surface wind but without any dust storms.

7. Section 2.3, why is the RAPv3 selected for the analysis? What variables are used?

8. Lines 129-130, which figure do you refer to?

9. Line 142, can you please provide definitions for 'blowing dust,' 'vicinity blowing dust,' and 'dust storm'?

10. Line 216, is there an upper limit of $PM_{10}$ measurement?

11. Line 265, is it the daily average of 26 Feb. 2023?

12. Line 280, how are correlations calculated? Do you use hourly data of the day? Are auto-correlations considered?

13. Line 285, are the correlations significant?

14. The current title indicates that meteorological conditions are affected by the dust storm. However, the text suggests the other way around. Please consider rewording to avoid confusion.

15. In Figure 2, please consider reducing the density of contours. In Fig. 2a-b, is the shading total wind speed? Also, can you please add labels for temperature and geopotential height? Similarly, for Fig. 2c, please add labels for surface height. And why are these time snapshots, i.e., 18UTC on 26 Feb. and 00UTC on 27 Feb., selected? When did the storm start?

---

## Author Comment (AC1)

Dear Editor,

Thank you for agreeing to consider a revision of our manuscript "The Spatial, Temporal, and Meteorological Impact of the 26 February 2023 Dust Storm, Increase of Particulate Matter Concentrations Across New Mexico and West Texas ". We modified and revised the manuscript to address the reviewers' comments as well as to clarify points that they found confusing or unclear.

We would like to thank the two reviewers, Dr. Allison C. Aiken and the anonymous reviewer for their helpful comments and suggestions, and many thanks to you for your time and efforts with this revision. In line with the comments and suggestions, we revised the manuscript and made the requested additions and changes. Below are all the comments (in bold) followed by the replies. The parts that are in italics are corrections that are included in the revised version of the paper:

Sincerely,
Karin Ardon-Dryer

**Review of egusphere-2024-113 by A.C. Aiken**
**Summary:**
**This manuscript describes a dust event that occurred in the desert southwestern United States. The data used includes meteorology and particulate matter (PM2.5 and PM10) from a range of available monitoring sites within New Mexico and Texas. The manuscript addresses relevant science questions within the scope of ACP, presenting data from 19 monitoring stations in NM and TX of a dust event that occurred in February 2023 and had the highest PM2.5 concentration recorded in the last 20 years at one of the sites (Lubbock, TX). Overall, the manuscript is well-written and worthy of publication in ACP. A few areas could be improved with minor revisions to highlight the importance of the results as they are presented in the current version to relate this dust event and storm within a larger context in terms of visibility, PM, and their impacts.**

**The authors' presentation of the data is well-structured and clear. The scientific impact could be improved by adding additional information in a general sense on how this event relates to others in region as well as the globe in the abstract and conclusion sections. Discussing a regional or seasonal average for visibility and PM versus those during the event would further highlight the magnitude of the different conditions experienced during the event versus the "background". Some more detail of how averages were calculated and reference periods with no dust events were selected within the methods section would be beneficial. Background conditions with no dust events are of interest and should be presented in the text as well as potentially in a table as the PM concentrations are hard to see in the figures since the event concentrations are so high in comparison. The conclusions section could be expanded as well, and more detailed suggestions are included in the comments below. For example, an interesting area to highlight would be a summary of the mass ratios for PM2.5/PM10 that were analyzed in the results section as well as a comparison of the PM data during the event versus the periods before and after the event to understand the magnitude of the impact on PM (and visibility). More details and recommendations are included below.**

We would like to thank the reviewer for the suggestions, corrections, and comments.

**General Comments:**
**Title**
**Clearly reflects the contents of the manuscript.**

The title of the manuscript was modified per the comments from both reviewers, this is the title of the revised manuscript:
*The Spatial, Temporal, and Meteorological Impact of the 26 February 2023 Dust Storm, Increase of Particulate Matter Concentrations Across New Mexico and West Texas*

**Abstract**
**Can you relate in general how this dust event and dust storm relate to others in the region and potentially globally in terms of duration, wind speeds, visibility, and PM concentrations?**

Since there is a number word limitation to the one that could be used in the abstract we could not add all the information requested by the reviewer, but we did add the information required in the main section of the manuscript. Please see the track changes document it will contain the many changes and information that was added per the reviewer's comments.

**It is highlighted that PM2.5 was the highest ever recorded at the Lubbock site. Could you also add to the abstract what the ratios of PM2.5/PM10 were on average during the event and storm as well as the dust storm in relation to the baseline or background and how this compares to other storms and/or regions?**

Per the reviewer's comment this information was added to the abstract, since there is a limitation on the number of words that could be used in the abstract, we only summarized the important information that needed to be highlighted. Detailed information is provided in the manuscript.

These changes were added to the revised manuscript abstract:
*The Southwestern portions of the United States experience dust events frequently due to the arid and semi-arid environments and close proximity to multiple deserts. On 26 February 2023, a dust event was initiated in New Mexico due to strong winds aloft mixing down to the surface. The dust intensified as it moved eastward into West Texas and developed into a dust storm (visibility < 1 km) for multiple locations. This study examined the meteorological characteristics of this dust storm using 28 meteorological stations and examined the impacts on $PM_{2.5}$ and/or $PM_{10}$ (particulate matter with an aerodynamic diameter < 10 and 2.5 μm) concentrations using 19 air quality stations. The dust event lasted up to ~16 hours, dust storm conditions lasted from five minutes up to 120 minutes. The highest wind speed and wind gust recorded during the dust were 27.3 m s$^{-1}$ and 37 m s$^{-1}$ respectively. This dust had a strong impact on the air quality in the area, as very high PM values were recorded across the region, and nine of the PM stations exceeded the EPA daily threshold. The maximum hourly $PM_{2.5}$ and $PM_{10}$ concentrations recorded were 518.4 μg m$^{-3}$ and 9,983 μg m$^{-3}$ respectively. These concentrations at the peak of the dust were an order of magnitude higher than the minimum hourly $PM_{2.5}$ and $PM_{10}$ concentrations recorded on*

*the dust day. The highest hourly PM$_{10}$-PM$_{2.5}$ concentration recorded was 759.8 $\pm$ 1000.3 $\mu g$ $m^{-3}$, while the lowest hourly PM$_{2.5}$/PM$_{10}$ concentration measured was 0.05 $\pm$ 0.01.*

**Introduction**
**Line 43: Recommendation to expand this paragraph on how this storm relates to others in the area. It is stated that this storm was "one of the most significant" over the last decade. Could a summary of the other storms be included to understand more of the climatology for the region? Are there other references to include to show historical comparisons for this area? Include how often they occur and last as well as available summaries/measurements of PM.**

**Is there a seasonality for the region and/or known occurrence in the area due to different meteorological patterns or events? There is some information on this provided in the results/paragraph Line 186 but adding a higher-level summary in the introduction would help the reader put this study within a larger context before getting into more of the details in the results section.**

Per the reviewer's two Introduction comments, additional information was added to the introduction in order to provide more information on dust in the southwestern U.S., particularly over New Mexico and western Texas. As well as PM information as requested by the reviewer. The last paragraph of this introduction was modified to reflect the reviewer's comment.

[revised manuscript text omitted]

**Methods**
**Add details of how the average daily values for PM were calculated. Was the dust event excluded or included and why? How much would those averages change if you did the opposite?**

The daily averages were calculated for each station based on hourly measurements from midnight to 23:00 LT. Since the dust happens on the 26[th] this is the day that represents the day with dust described in this paper, but we also present concentrations during the time of the dust (which are based on PM increase). Per the reviewer's comments, we made different calculations including

only the dime of dust, the peak of dust which is reported hourly maximum, daily concentrations, and as well per some of the following comments by the reviewers monthly average and average for every day during the same month.

This information on that calculation was added to the methods section.:

*Calculations of the daily average were made for each day based on hourly measurements from midnight to 23:00 local time (LT).*

**If the dust event was included in the daily averages should the day before and after be included to assess how much the daily average changed during the day of the event?**

The main reason for using the daily average (midnight to 23:00) was based on the fact that in order to compare to other studies and to examine air quality aspects, based on EPA and WHO, daily values need to be calculated. We do not think that just showing the day before or the day after would have been enough therefore we added more information to the revised manuscript per several of the reviewers' comments about this topic. We first provided the PM concentration during the time of dust and compared that to the daily average, since the dust did not last 24 hours in any of the locations examined. Next, we examine the daily average for each one of the days during the month (as suggested by the reviewer in the following comment). We added a figure to the supplant that shows these daily changes. Most figures highlight the impact of the examined dust event compared to the additional days of the month, but it should be noted that there was additional dust events during that month, but they were not as strong as the one presented in this study.

This information was added to the revised manuscript:
*To examine the impact of the 26 February 2023 dust event on the overall PM concentrations, daily PM concentrations were calculated, for each PM sensor, for each day during February 2023 (shown in Fig. S2). The daily average for February 26 seems high (for most sensors) compared to the other February day's daily average, it also seems to have much higher SD values compared to manty of the other days. The lowest impact seems to be in the Albuquerque stations, perhaps since the area is also impacted by anthropogenic pollution. The southern part of New Mexico and many of the stations in West Texas seem to have had a bigger impact on this dust event, as daily values for the dust day (February 26) were on average 12 times higher compared to the overall daily $PM_{2.5}$ concentrations and 27.5 times higher compared to the $PM_{10}$ daily concentrations. These differences could have been higher, but it seems there were additional pollution events (other dust events, as indicated above) in some of the locations, which increased the daily PM concentration for some days in some of the stations. Observations of daily $PM_{2.5}$ concentrations from the different Albuquerque PM stations show that the dust was not as strong as it was for other locations such as South New Mexico and West Texas. Next, the monthly $PM_{2.5}$ and $PM_{10}$ concentrations for the entire February month were calculated (for each sensor), without February 26 PM concentrations (Table S4). The monthly $PM_{2.5}$ and $PM_{10}$ concentrations were on average 4.0 and 9.3 times lower, for $PM_{2.5}$ and $PM_{10}$ concentrations respectively, compared to the daily concentrations measured on February 26. The monthly PM concentrations were 13.6 and 26.7 times lower compared to the $PM_{2.5}$ and $PM_{10}$ concentrations (respectively) measured during the time of the dust, and 26.4 and 104.9 times lower (for $PM_{2.5}$ and $PM_{10}$ concentrations, respectively) compared to the PM*

*concentrations at the peak of the dust. These large differences between the concentrations of PM$_{2.5}$ and PM$_{10}$ during the dust to those over the month indicate that while the background PM across the region might be low (except for Albuquerque) dust events in this region can have a significant impact on both PM$_{2.5}$ and PM$_{10}$ concentrations in the region which will impact on the air quality wellbeing and peoples health.*

[Figure]

*Figure S2. Changes in a daily average of PM$_{2.5}$ (black) and PM$_{10}$ (blue) every day during February 2023 with SD values for each day. The daily average for February 26 is presented in orange.*

**When concentration ratios are calculated for the event versus the period before – how was the period before defined? How would it compare to an average after the event – do the concentrations return to the previous values or is there a significant difference or lag in time before the values return to the pre-event concentrations?**

Per the reviewer's comment, we provided in the revised manuscript additional explanation and calculation. First, one difference was between the peak of the dust to the concentration right before the dust arrived at the station. We also examined the difference between the concentration at the

peak of the dust to the minimum concentrations recorded during the dust day by each of the stations. We also provide an examination of the daily average concentration during the dust day to other days during this month and to a monthly average concentration (excluding the dust day from that calculation).

This information was added to the revised manuscript:

*The duration when dust particles were in the air based on an increase in PM values was similar to the duration based on visibility, mentioned in section 3.2. These durations based on PM values varied, some stations had an increase in PM values for a duration of 2 hours, while others for up to 12 hours. Despite the reduced visibility to 1.6 km during the dust event at the Albuquerque ASOS (ABQ), most of the PM stations in the area witnessed a small increase in $PM_{2.5}$ but a more significant increase in $PM_{10}$ concentrations (as can be seen in Fig. 5 and Table S4). A spatial impact of the dust was also observed in Albuquerque, as stations in the southern part of Albuquerque had higher PM concentrations (with a stronger increase) compared to those located in the northern part of Albuquerque. When calculating the increased ratio of PM, which is indicated by the ratio of PM concentrations at the peak of the dust compared to the PM concentrations right before the dust, results showed an increase in PM across the region, even across Albuquerque. $PM_{2.5}$ concentrations during the dust event were on average 12.8 times higher compared to the time before the dust event (ratios vary from 3.0 up to 36.3), while $PM_{10}$ concentrations during the dust event were on average 216.9 times higher compared to before the dust event (ratios vary from 11.3 up to 1426.1). When we examined the same ratio aspect for the lowest PM concentrations recorded on February 26 (shown in Table S4) the differences were much higher, $PM_{2.5}$ and $PM_{10}$ concentrations at the peak of the dust were higher by more than an order of magnitude (on average) than the minimum daily PM concentration recorded on the same day.*

**Would it be possible to look at a longer background period such as a week or a month to establish a "background" for the data?**

Per the reviewer's comment, we provided in the revised manuscript a comparison between the dust time and day for the entire month of February 2023 as well as all daily averages measured during this month by each of the stations. This information was added to the supplemental section as a table (Table S3) and also as a figure. The information was added for both PM concentrations but also the meteorological information, as well as described in the following comment.

**Results**
**Would it be possible to calculate background values for visibility and PM over a longer time period and to discuss what those values are for a month or the season?**

Per the reviewer's comment, we downloaded again the ASOS data from the entire month of February 2023, and the PM data (as mentioned in previous comments). In the process, we found additional ASOS stations that could be used to highlight the findings of our work, and we added them to the revised manuscript which explains the changes performed to the figures. Following this reviewer, we calculated the monthly wind speed and visibility values excluding the February

23 dust events to highlight the impact of these dust events on these parameters. This information was added to the supplemental section (Table S3).

This information was added to the revised manuscript:

*These wind speeds and wind gusts measured during the dust events were 3.2 times higher than the average wind speed and wind gust recorded in the month of February 2023 (shown in Table S3). The difference was much stronger for the strongest recorded wind speed and wind gust, up to 5.9 times and 8.3 times (respectively) compared to the month of February. These big differences indicate how strong this dust event was. But looking at the overall meteorological conditions during this month, it seems that there were additional dust events during that month (e.g., February 9 and 22), but were not as strong as the one reported here (data not shown). Perhaps if these dust times had been removed from the monthly analysis the difference between the meteorological conditions would have been stronger.*

*Next, the monthly $PM_{2.5}$ and $PM_{10}$ concentrations for the entire February month were calculated (for each sensor), without February 26 PM concentrations (Table S4). The monthly $PM_{2.5}$ and $PM_{10}$ concentrations were on average 4.0 and 9.3 times lower, for $PM_{2.5}$ and $PM_{10}$ concentrations respectively, compared to the daily concentrations measured on February 26. The monthly PM concentrations were 13.6 and 26.7 times lower compared to the $PM_{2.5}$ and $PM_{10}$ concentrations (respectively) measured during the time of the dust, and 26.4 and 104.9 times lower (for $PM_{2.5}$ and $PM_{10}$ concentrations, respectively) compared to the PM concentrations at the peak of the dust. These large differences between the concentrations of $PM_{2.5}$ and $PM_{10}$ during the dust to those over the month indicate that while the background PM across the region might be low (except for Albuquerque) dust events in this region can have a significant impact on both $PM_{2.5}$ and $PM_{10}$ concentrations in the region which will impact on the air quality wellbeing and peoples health.*

**It looks like visibility may be more stable than PM, but it's not easy to tell from the figures in the main text what a non-dust period looks like in terms of PM.**

We believe the review came to this conclusion since the original PM figure aims to keep all the scales of PM at the same level. Per the reviewer's comment, we modified the PM figure to highlight the impact of PM in each station and figure.

This is the new figure in the revised manuscript

[Figure]

*Figure 5. Changes in PM$_{2.5}$ (black) and PM$_{10}$ (blue) with wind speeds (grey) measured during the dust storm. The name of the station and daily average ± SD values for February 26 are presented in black. daily average ± SD values for stations that exceeded the EPA daily standards are presented in red.*

**If a month is too long, even a week would show a longer period to give an idea of PM and ratios between the size cuts for the region. Ideally, you could look at all the station averages as well as an average for all the data presented. For example, there look to be two main areas where the sites have both PM10 and PM2.5. How similar/different are the concentrations and ratios at those two sites and what might be the reasons for those differences that warrant further investigation.**

Per the reviewer's comment, we downloaded again the PM data from the entire month of February 2023 (as mentioned in previous comments). We did not want to combine data between stations as each reflects a different location, they were also far from each other. Per this comment, we added a new table to the supplement that provides the ratio and values of $PM_{2.5}/PM_{10}$ and $PM_{10}$-$PM_{2.5}$. Below is a picture of that table it can be found in the supplement file.

*Table S5. Measurements of $PM_{10}$-$PM_{2.5}$ and $PM_{2.5}/PM_{10}$ during the 26 February dust storm and the month of February 2023.*

| PM | Location | Station ID | $PM_{10}$-$PM_{2.5}$ (µg m -3) | | | | $PM_{2.5}/PM_{10}$ (µg m -3) | | | |
|---|---|---|---|---|---|---|---|---|---|---|
| | | | Daily (02/26) average Average ± SD | Average during time of dust Average ± SD | Max Peak of dust | Monthly average (without 02/26) Average ± SD | Daily (02/26) average Average ± SD | Average during time of dust Average ± SD | Min Peak of dust | Monthly average (without 02/26) Average ± SD |
| Stations with $PM_{10}$ & $PM_{2.5}$ | El Paso | C49 | 101 ± 218 | 451.7 ± 325.4 | 870.5 | 28.7 ± 20.7 | 0.25 ± 0.16 | 0.09 ± 0.0 | 0.02 | 0.19 ± 0.08 |
| | El Paso | C41 | 49.6 ± 88.3 | 186.6 ± 146.1 | 503 | 22.4 ± 20.3 | 0.26 ± 0.35 | 0.07 ± 0.0 | 0.03 | 0.43 ± 0.24 |
| | Anthony | 6CM | 159 ± 508 | 759.8 ± #### | 3236 | 40.7 ± 53.5 | 0.24 ± 0.24 | 0.05 ± 0.0 | 0.03 | 0.10 ± 0.08 |
| | Albuquerque | Del Norte HS | 19.9 ± 39.8 | 114.0 ± 106.2 | 270.9 | 12.3 ± 7.2 | 0.28 ± 0.12 | 0.09 ± 0.4 | 0.06 | 0.31 ± 0.07 |
| | Albuquerque | Foothills | 15 ± 29.5 | 95.9 ± 61.3 | 183.9 | 9.1 ± 3.8 | 0.26 ± 0.10 | 0.08 ± 0.3 | 0.07 | 0.31 ± 0.07 |
| | Albuquerque | Jefferson | 44.4 ± 135 | 309.6 ± 409.2 | 922.7 | 20.8 ± 19.5 | 0.25 ± 0.10 | 0.08 ± 0.3 | 0.06 | 0.27 ± 0.07 |
| | Albuquerque | North Valley | 26 ± 50.9 | 161.2 ± 115.4 | 330.2 | 18.4 ± 12.7 | 0.3 ± 0.14 | 0.06 ± 0.2 | 0.05 | 0.32 ± 0.08 |
| | Albuquerque | San Jose | 41.7 ± 137 | 279.2 ± 374.0 | 942.7 | 90.4 ± 35.2 | 0.27 ± 0.12 | 0.06 ± 0.3 | 0.05 | 0.14 ± 0.08 |
| | Albuquerque | South Valley | 71.1 ± 298 | 518.4 ± 848.0 | 2030.9 | 28.7 ± 30.1 | 0.24 ± 0.12 | 0.07 ± 0.3 | 0.05 | 0.27 ± 0.08 |

**Could you add a table to the main text that shows the background values for visibility and PM versus the event average and the peak during the event?**

The information requested by the reviewer for both PM and visibility was added to the revised manuscript as a supplemental table. Table S3 provides a comparison to the monthly meteorological conditions (wind speed and visibility), Table S4 provides a comparison to the monthly PM concentration and Table S5 provides a comparison to the monthly $PM_{2.5}/PM_{10}$ and $PM_{10}$-$PM_{2.5}$ concentration. Since these two are big for the Word document they are provided as pictures below. The table can be found in the supplement file.

*Table S3. Meteorological parameters measured by ASOS stations during the dust event. Duration of the dust storm (DS) reported only for stations that reported visibility below 1 km. Bold numbers represent stations with visibilities < 1 km therefore DS.*

| | Station ID | Start Time of dust (local time) first VIS <10 km | End Time (local time) of dust first VIS >10 km | Duration of Dust Event (hour) VIS <10 km | Wind Speed at Start (m s-1) | Wind Gust at Start (m s-1) | Max Wind Speed During Dust Event (m s-1) | Max Wind Gust During Dust Event (m s-1) | Lowest Recorded Visibility (km)* | Duration of DS | Wind Speed during dust Average ± SD (m s-1) | Visibility during dust times Average ± SD (m s-1) | Wind Gust during dust Average ± SD (m s-1) | Wind Speed during Feb 2023 (excluding dust times) Average ± SD (m s-1) | Visibility during Feb 2023 (excluding dust times) Average ± SD (m s-1) |
|---|---|---|---|---|---|---|---|---|---|---|---|---|---|---|---|
| New Mexico | 0E0 | 12:55 | 14:35 | 1:40 | 18.5 | 26.7 | 18.5 | 26.7 | 2.4 | | 15.9 ± 1.7 | 3.5 ± 0.9 | 22.1 ± 3.2 | 4.7 ± 3.3 | 15.5 ± 2.6 |
| | ABQ | 10:21 | 13:20 | 2:59 | 17.0 | 22.1 | 23.6 | 33.4 | 1.6 | | 15.7 ± 3.9 | 7.6 ± 4.0 | 22.4 ± 5.6 | 4.0 ± 2.8 | 15.6 ± 2.3 |
| | ALM | 11:55 | 18:15 | 6:20 | 18.5 | 25.2 | 20.0 | 26.2 | **0.4** | 0:20 | 14.4 ± 3.4 | 3.9 ± 2.1 | 19.9 ± 4.1 | 4.4 ± 2.4 | 16.0 ± 0.9 |
| | ATS | 13:15 | 15:55 | 2:40 | 20.6 | 30.3 | 24.7 | 31.9 | 2.8 | | 21.7 ± 1.5 | 6.5 ± 2.6 | 28.9 ± 2.2 | 6.1 ± 4.1 | 15.7 ± 2.2 |
| | CNM | 13:35 | 16:05 | 2:30 | 20.0 | 24.7 | 23.1 | 31.9 | 4.0 | | 19.4 ± 1.9 | 7.4 ± 2.2 | 25.3 ± 3.4 | 5.5 ± 3.8 | 15.9 ± 1.4 |
| | CVN | 12:25 | 18:15 | 5:50 | 15.4 | 19.5 | 25.7 | 31.9 | **0.4** | 1:10 | 20.7 ± 2.3 | 3.1 ± 2.9 | 25.5 ± 2.9 | 6.7 ± 3.8 | 15.7 ± 1.8 |
| | DMN | 10:00 | 12:20 | 2:20 | 13.4 | 19.0 | 22.1 | 30.3 | **0.4** | 0:30 | 15.2 ± 2.4 | 9.2 ± 5.4 | 21.0 ± 3.2 | 4.8 ± 3.2 | 15.9 ± 1.4 |
| | HMN | 12:17 | 18:44 | 6:27 | 21.1 | 29.3 | 25.2 | 29.3 | **0.8** | 0:09 | 16.8 ± 4.2 | 3.8 ± 2.2 | 23.1 ± 4.5 | 4.8 ± 2.9 | 16.0 ± 0.9 |
| | LRU | 10:55 | 16:35 | 5:40 | 18.0 | 24.7 | 23.6 | 31.9 | 1.2 | | 19.7 ± 2.1 | 4.9 ± 2.3 | 26.8 ± 2.3 | 4.7 ± 3.2 | 16.0 ± 0.8 |
| | ROW | 12:45 | 18:33 | 5:48 | 19.5 | 24.2 | 25.7 | 35.0 | **0.8** | 0:10 | 17.2 ± 4.4 | 6.1 ± 4.0 | 23.9 ± 5.9 | 4.5 ± 3.2 | 15.9 ± 0.9 |
| | SRR | 12:55 | 15:55 | 3:00 | 25.7 | 33.4 | 25.7 | 33.4 | 1.2 | | 15.4 ± 5.6 | 3.2 ± 1.7 | 20.9 ± 7.9 | 6.0 ± 3.9 | 15.5 ± 2.5 |
| | TCC | 12:30 | 16:45 | 4:15 | 19.5 | 27.8 | 27.2 | 37.0 | **0.8** | 0:20 | 20.3 ± 2.4 | 6.4 ± 4.2 | 27.6 ± 4.0 | 6.2 ± 4.1 | 15.9 ± 1.5 |
| Texas | AMA | 14:50 | 22:38 | 7:48 | 18.5 | 22.1 | 25.2 | 33.4 | 1.2 | | 18.2 ± 2.5 | 4.7 ± 3.0 | 25.1 ± 3.7 | 6.7 ± 3.3 | 15.9 ± 1.4 |
| | BIF | 11:55 | 17:55 | 6:00 | 17.0 | 21.1 | 17.0 | 26.7 | 3.2 | | 14.3 ± 2.1 | 4.6 ± 1.7 | 21.9 ± 2.3 | 4.9 ± 3.7 | 15.5 ± 1.6 |
| | BPG | 16:35 | 23:55 | 7:20 | 14.4 | 20.6 | 16.5 | 21.1 | 2.4 | | 11.9 ± 3.1 | 6.3 ± 4.1 | 16.8 ± 2.6 | 5.8 ± 3.1 | 15.3 ± 2.6 |
| | E11 | 16:15 | 19:15 | 3:00 | 18.0 | 23.6 | 18.5 | 25.2 | 1.5 | | 16.3 ± 1.8 | 6.7 ± 1.5 | 22.8 ± 1.7 | 5.0 ± 2.7 | 15.7 ± 2.2 |
| | ELP | 11:10 | 14:15 | 3:05 | 17.0 | 21.6 | 20.6 | 31.4 | 2.4 | | 17.3 ± 2.0 | 5.6 ± 2.5 | 23.6 ± 3.1 | 4.4 ± 3.2 | 16.0 ± 0.9 |
| | GNC | 15:50 | 22:50 | 7:00 | 14.4 | 18.5 | 19.5 | 25.7 | 1.6 | | 14.3 ± 2.9 | 6.5 ± 4.3 | 20.5 ± 3.8 | 5.0 ± 2.9 | 15.8 ± 1.9 |
| | HRX | 14:55 | 21:55 | 7:00 | 17.0 | 23.1 | 26.2 | 30.3 | **0.4** | 1:20 | 19.0 ± 3.1 | 3.2 ± 2.3 | 25.3 ± 3.5 | 6.3 ± 3.6 | 15.9 ± 1.1 |
| | INK | 14:53 | 19:25 | 4:32 | 15.4 | 24.7 | 23.1 | 31.9 | **0.8** | 0:05 | 18.4 ± 2.3 | 3.3 ± 1.9 | 24.7 ± 3.8 | 4.6 ± 2.9 | 15.7 ± 1.9 |
| | LBB | 14:53 | 3:20 | 12:27 | 14.9 | 21.1 | 25.7 | 33.9 | **0.0** | 1:55 | 17.7 ± 2.8 | 4.8 ± 2.7 | 23.1 ± 4.2 | 6.4 ± 3.5 | 15.4 ± 2.4 |
| | LLN | 14:35 | 4:55 | 14:20 | 14.4 | 18.5 | 25.7 | 31.4 | **0.4** | 2:00 | 16.3 ± 3.1 | 3.9 ± 2.6 | 22.0 ± 4.2 | 6.7 ± 3.6 | 15.2 ± 2.6 |
| | LUV | 12:35 | 3:55 | 15:20 | 13.4 | 17.0 | 20.0 | 24.7 | **0.8** | 1:00 | 14.5 ± 2.1 | 5.0 ± 3.0 | 19.0 ± 2.9 | 6.0 ± 3.1 | 15.4 ± 2.6 |
| | MAF | 17:00 | 20:25 | 3:25 | 12.3 | 15.9 | 20.6 | 28.8 | 2.0 | | 15.6 ± 2.6 | 6.4 ± 2.6 | 20.9 ± 3.4 | 5.3 ± 3.0 | 14.0 ± 3.1 |
| | MDD | 18:35 | 19:55 | 1:20 | 17.5 | 22.6 | 17.5 | 22.6 | 3.2 | | 15.3 ± 1.5 | 5.6 ± 2.1 | 19.9 ± 1.9 | 5.1 ± 2.6 | 15.6 ± 2.1 |
| | ODD | 17:15 | 20:00 | 2:45 | 14.9 | 23.1 | 18.5 | 25.7 | 2.0 | | 15.2 ± 1.9 | 5.5 ± 2.7 | 21.2 ± 2.8 | 5.1 ± 2.8 | 15.0 ± 3.0 |
| | PEQ | 13:55 | 19:35 | 5:40 | 14.4 | 20.6 | 20.0 | 25.7 | 1.2 | | 16.3 ± 2.3 | 5.8 ± 4.4 | 22.0 ± 2.2 | 4.7 ± 3.1 | 15.8 ± 1.4 |
| | VHN | 12:15 | 18:35 | 6:20 | 12.9 | 20.6 | 17.5 | 25.7 | 1.6 | | 14.5 ± 2.0 | 6.2 ± 3.8 | 20.4 ± 2.5 | 4.4 ± 2.2 | 16.0 ± 1.1 |

\* Bold values represent Visibility < 10 km

Table S4. Measurements of $PM_{2.5}$ and $PM_{10}$ during the 26 February dust storm and the month of February 2023. Bold numbers represent significant $R^2$ values.

| PM | Location | Station ID | PM values average during time of dust (µg m-3) | | | | R² for hourly wind speed & PM * | | Daily (02/26) average (µg m-3) | | Feb Monthly average (µg m-3) without 02/26 |
|---|---|---|---|---|---|---|---|---|---|---|---|
| | | | Average ± SD | Max Peak of dust | Number of hours with high PM | Increase ratio of PM+ | Linear reg. | Polynomial reg. | Daily Min | Average ± SD | Average ± SD |
| PM2.5 | Amarillo | C320 | 93.7 ± 42.4 | 161.0 | 6 | 8.05 | 0.41 | **0.57** | 9.0 | 36 ± 39.9 | 3.8 ± 2.3 |
| | Lubbock | C1028 | 153.9 ± 134.7 | 518.4 | 12 | 22.6 | 0.15 | 0.18 | 3.9 | 69.3 ± 121.3 | 5.8 ± 7.3 |
| | Odessa | C1014 | 74.5 ± 38.9 | 102.0 | 2 | 3.9 | 0.28 | **0.51** | 5.0 | 18.3 ± 19.8 | 5.8 ± 3.6 |
| | El Paso | C37 | 147.6 ± 113.8 | 346.2 | 5 | 17.2 | 0.25 | 0.4 | 14.0 | 53.6 ± 69 | 10.9 ± 6.0 |
| | Hobbs | 5ZS | 99.7 ± 47.1 | 142.0 | 3 | 11.8 | 0.47 | **0.77** | 2.0 | 21.1 ± 33.7 | 2.5 ± 2.0 |
| | Las Cruces | 6Q | 107.0 ± 28.3 | 127.0 | 2 | 21.2 | 0.37 | **0.7** | 2.0 | 18.8 ± 28.8 | 4.4 ± 1.4 |
| | El Paso | C49 | 79.2 ± 46.8 | 132.6 | 4 | 4.42 | 0.47 | **0.88** | 3.2 | 22.3 ± 35.8 | 5.5 ± 2.2 |
| | El Paso | C41 | 76.8 ± 35.5 | 109.0 | 4 | 36.3 | 0.27 | **0.71** | 0.0 | 19.8 ± 29.6 | 9.1 ± 5.9 |
| | Anthony | 6CM | 104.3 ± 117.1 | 239.0 | 3 | 11.8 | 0.34 | **0.79** | 0.0 | 20.5 ± 47.8 | 4.2 ± 3.4 |
| | Albuquerque | Del Norte HS | 9.6 ± 6.2 | 18.7 | 4 | 3.2 | 0.02 | **0.73** | 1.1 | 6.7 ± 5.8 | 5.1 ± 2.0 |
| | Albuquerque | Foothills | 8.4 ± 4.0 | 13.6 | 4 | 3.8 | 0.22 | 0.23 | 0.6 | 3.8 ± 2.7 | 4.0 ± 1.1 |
| | Albuquerque | Jefferson | 23.0 ± 26.6 | 62.8 | 4 | 9 | 0.16 | **0.8** | 1.1 | 8.8 ± 12.9 | 6.4 ± 2.3 |
| | Albuquerque | North Valley | 9.6 ± 4.9 | 16.6 | 4 | 3 | 0.01 | 0.49 | 1.9 | 7 ± 4.4 | 7.9 ± 3.2 |
| | Albuquerque | San Jose | 18.3 ± 18.3 | 45.6 | 4 | 7.7 | 0.14 | **0.9** | 1.3 | 7.5 ± 8.9 | 12.6 ± 5.7 |
| | Albuquerque | South Valley | 37.4 ± 50.0 | 112.3 | 4 | 9.9 | 0.14 | **0.9** | 1.0 | 13.6 ± 22.1 | 8.7 ± 3.9 |
| PM10 | El Paso | C49 | 461.4 ± 371.5 | 999.4 | 10 | 44.4 | **0.6** | **0.86** | 9.9 | 205.1 ± 320.9 | 34.3 ± 22.0 |
| | El Paso | C41 | 204.1 ± 168.5 | 572.0 | 10 | 11.9 | 0.4 | **0.73** | 6.0 | 88.4 ± 114.6 | 26.9 ± 25.4 |
| | Chaparral | 6ZK | 577.1 ± 643.0 | 2254.0 | 11 | 93.9 | 0.35 | 0.44 | 2.0 | 278.2 ± 508.6 | 24.3 ± 27.2 |
| | Desert View | 6ZM | 1752.6 ± ##### | 9983.0 | 10 | 1426.1 | 0.4 | **0.72** | 2.0 | 748.4 ± 2089.7 | 48.8 ± 46.6 |
| | Las Cruces | 6ZL | 2353.7 ± ##### | 9868.0 | 7 | 704.9 | 0.36 | **0.7** | 2.0 | 700.8 ± 2198.5 | 21.4 ± 31.2 |
| | Anthony | 6CM | 803.1 ± ##### | 3475.0 | 9 | 119.8 | **0.52** | **0.87** | 2.0 | 323.2 ± 741.9 | 44.9 ± 55.9 |
| | Las Cruces | 6WM | 1212.1 ± ##### | 4469.0 | 8 | 235.2 | 0.42 | **0.67** | 0.0 | 415 ± 1065.8 | 13.7 ± 19.7 |
| | Albuquerque | Del Norte HS | 123.6 ± 112.3 | 289.6 | 4 | 13.3 | 0.33 | **0.9** | 4.2 | 37.1 ± 57.8 | 17.4 ± 8.0 |
| | Albuquerque | Foothills | 104.3 ± 65.2 | 197.5 | 4 | 11.3 | 0.33 | 0.37 | 2.9 | 28.1 ± 42.3 | 13.1 ± 4.4 |
| | Albuquerque | Jefferson | 332.5 ± 435.8 | 985.5 | 4 | 25.5 | 0.3 | **0.87** | 4.3 | 76.7 ± 199.3 | 27.1 ± 20.4 |
| | Albuquerque | North Valley | 170.8 ± 120.3 | 346.8 | 4 | 15.9 | **0.5** | **0.86** | 4.6 | 86.3 ± 199.8 | 26.3 ± 13.8 |
| | Albuquerque | San Jose | 357.1 ± 421.8 | 988.3 | 4 | 45.1 | 0.36 | **0.88** | 5.6 | 46.8 ± 72.1 | 103.0 ± 38.1 |
| | Albuquerque | South Valley | 469.5 ± 822.8 | 2143.2 | 6 | 72.4 | 0.31 | **0.86** | 5.0 | 136.7 ± 431.2 | 37.4 ± 31.8 |

+ Increase the ratio of PM representing the ratio between the peak of dust to PM measurements right before dust made it to the station

\* Bold values represent cases with $R^2$ values > 0.05.

*Table S5. Measurements of PM$_{10}$-PM$_{2.5}$ and PM$_{2.5}$/PM$_{10}$ during the 26 February dust storm and the month of February 2023.*

| PM | Location | Station ID | PM$_{10}$-PM$_{2.5}$ (µg m -3) | | | | PM$_{2.5}$/PM$_{10}$ (µg m -3) | | | |
|---|---|---|---|---|---|---|---|---|---|---|
| | | | Daily (02/26) average | Average during time of dust | Max Peak of dust | Monthly average (without 02/26) | Daily (02/26) average | Average during time of dust | Min Peak of dust | Monthly average (without 02/26) |
| | | | Average ± SD | Average ± SD | | Average ± SD | Average ± SD | Average ± SD | | Average ± SD |
| Stations with PM$_{10}$ & PM$_{2.5}$ | El Paso | C49 | 101 ± 218 | 451.7 ± 325.4 | 870.5 | 28.7 ± 20.7 | 0.25 ± 0.16 | 0.09 ± 0.0 | 0.02 | 0.19 ± 0.08 |
| | El Paso | C41 | 49.6 ± 88.3 | 186.6 ± 146.1 | 503 | 22.4 ± 20.3 | 0.26 ± 0.35 | 0.07 ± 0.0 | 0.03 | 0.43 ± 0.24 |
| | Anthony | 6CM | 159 ± 508 | 759.8 ± #### | 3236 | 40.7 ± 53.5 | 0.24 ± 0.24 | 0.05 ± 0.0 | 0.03 | 0.10 ± 0.08 |
| | Albuquerque | Del Norte HS | 19.9 ± 39.8 | 114.0 ± 106.2 | 270.9 | 12.3 ± 7.2 | 0.28 ± 0.12 | 0.09 ± 0.4 | 0.06 | 0.31 ± 0.07 |
| | Albuquerque | Foothills | 15 ± 29.5 | 95.9 ± 61.3 | 183.9 | 9.1 ± 3.8 | 0.26 ± 0.10 | 0.08 ± 0.3 | 0.07 | 0.31 ± 0.07 |
| | Albuquerque | Jefferson | 44.4 ± 135 | 309.6 ± 409.2 | 922.7 | 20.8 ± 19.5 | 0.25 ± 0.10 | 0.08 ± 0.3 | 0.06 | 0.27 ± 0.07 |
| | Albuquerque | North Valley | 26 ± 50.9 | 161.2 ± 115.4 | 330.2 | 18.4 ± 12.7 | 0.3 ± 0.14 | 0.06 ± 0.2 | 0.05 | 0.32 ± 0.08 |
| | Albuquerque | San Jose | 41.7 ± 137 | 279.2 ± 374.0 | 942.7 | 90.4 ± 35.2 | 0.27 ± 0.12 | 0.06 ± 0.3 | 0.05 | 0.14 ± 0.08 |
| | Albuquerque | South Valley | 71.1 ± 298 | 518.4 ± 848.0 | 2030.9 | 28.7 ± 30.1 | 0.24 ± 0.12 | 0.07 ± 0.3 | 0.05 | 0.27 ± 0.08 |

**Consider moving the figures within S4 that contain the differences and ratios between PM10 and PM2.5 into the main text and expanding the discussion.**

The figure was moved to the main manuscript per the reviewer's comment, we also added more discussion on the matter and also comparison between our findings to those of other papers.

This information was added to the revised manuscript:

*The hourly PM$_{10}$-PM$_{2.5}$ values from this study were higher, for most stations, compared to values measured in three different dust events in Lubbock Texas, perhaps because this dust event was stronger (Ardon-Dryer and Kelley, 2022). The PM$_{10}$-PM$_{2.5}$ values were higher than those reported in the Rocky Mountains (Reynold et al., 2016) and Utah (Hahnenberger and Nicoll, 2012). Similar ranges of PM$_{10}$-PM$_{2.5}$ values were measured during dust storms in Israel (Krasnov et al., 2016). The daily values were lower compared to those measured in Israel, although the values at the peak of the dust were in the same range (Ardon-Dryer and Levin, 2014). However, the peak PM$_{10}$-PM$_{2.5}$ values were lower compared to the average PM$_{10}$-PM$_{2.5}$ values measured during multiple dust storms in China (Shao and Mao, 2016). Daily PM$_{10}$-PM$_{2.5}$ values in this dust event (for some of the stations) were in a similar range to those measured by Tong et al (2012), who examined multiple dust events in the same area as the one in this study.*

*Observations based on PM$_{2.5}$/PM$_{10}$ were also performed. The PM$_{2.5}$/PM$_{10}$ ratio is an important indicator used to characterize the underlying atmospheric processes within the local environment, which allows for the identification of the source of the particles (Yu and Wang, 2010). Higher PM$_{2.5}$/PM$_{10}$ ratios (> 0.6) are generally associated with anthropogenic pollution, while lower ratios are associated with dust events (Jugder et al., 2014; Sugimoto et al., 2016; Jaafari et al., 2018; Fan et al., 2021; Ardon-Dryer et al., 2022b). PM$_{2.5}$/PM$_{10}$ values across the nine sensors decreased during the dust event mainly between 11:00 to 18:00 LT (Fig. 6). PM$_{2.5}$/PM$_{10}$ values across the nine stations ranged from 0.03 to 0.13 with an average of 0.07 ± 0.02 across all stations and times. These ratios were lower compared to the values reported by Tong et al. (2012), which were 0.22-0.24, for this area combined with multiple dust events. Since Tong et al. (2012)*

*$PM_{2.5}/PM_{10}$ values were based on daily values calculations of daily values for each sensor were made (Table S5). The daily $PM_{2.5}/PM_{10}$ values were in the same range (and even slightly higher, 0.24 -0.3) as those in Tong et al. (2012). However, observations of these ratios during the time of dust (which were shorter than the duration of the day, as discussed above) were lower, with the average $PM_{2.5}/PM_{10}$ value of 0.07 (values across all stations ranged from 0.05 to 0.09). These values were similar to those measured by Li et al. (2005) during dust events in the El Paso region. The hourly $PM_{2.5}/PM_{10}$ values at the peak of the dust were lower compared to those measured at the peak in multiple dust storms in Utah (Hahnenberger and Nicoll, 2012; Nicoll et al., 2020). In Washington state, a similar range of daily $PM_{2.5}/PM_{10}$ values was measured (Claiborn et al., 2000). The daily $PM_{2.5}/PM_{10}$ values were in a similar range to those measured during dust events around the world (Alghamdi et al., 2015; Malaguti et al., 2015; Sugimoto et al., 2016; Jaafari et al., 2018).*

**Would it be possible to extend the PM2.5/PM10 ratio figure beyond the day of the event to look at that ratio in a broader context?**

We did not want to confuse the reader with observations over a longer period, mainly as previous days during the month experienced dust. We wanted the focus to be on these dust events. We believe the extension of the figure allows us to see how the ratio of $PM_{2.5}/PM_{10}$ and $PM_{10}$-$PM_{2.5}$ changes over time during the day of dust and the information we added about the monthly values provided allows us to get a broader context of the ratio of the dust event presented. Yet we do provide additional information as the average of $PM_{2.5}/PM_{10}$ and $PM_{10}$-$PM_{2.5}$ during the time of dust, as well as the hourly peak, daily average as well as monthly average which show how different these ratios were compared to those measured during the dust.

This information was added to the revised manuscript:

*Daily $PM_{10}$-$PM_{2.5}$ for February 26 were lower (3.8 up to 7.3 times, average of 5.8) compared to the $PM_{10}$-$PM_{2.5}$ calculated during the time of dust (Table S5). Also, $PM_{10}$-$PM_{2.5}$ at the peak of the dust was 16.6 times higher compared to the daily values. Calculations based on $PM_{10}$-$PM_{2.5}$ for each station for February showed that most stations had a low impact of coarse particles (except for San Jose and 6CM, which had higher monthly $PM_{10}$-$PM_{2.5}$ values, most likely due to the other dust events earlier that month). Both the $PM_{10}$-$PM_{2.5}$ values during the time of dust were higher (3.1 to 18.7 times, 11.9 on average) than the $PM_{10}$-$PM_{2.5}$ monthly values.*

*Observations of the monthly $PM_{2.5}/PM_{10}$ values for February (without the February 26 day, shown in Table S5), ranged from 0.1 ± 0.08 (for 6CM) up to 0.43 ± 0.24 (for C41). Most of the stations had lower monthly values compared to the daily $PM_{2.5}/PM_{10}$ values, some stations had similar values of ~1. The February 26 daily $PM_{2.5}/PM_{10}$ values were on average 3.6 times lower than the monthly values while the $PM_{2.5}/PM_{10}$ values at the peak of the dust were on average 6.2 times lower. The difference was slightly higher when monthly $PM_{2.5}/PM_{10}$ values were calculated without all the other suspected dust events (as mentioned in section 3.2).*

**Are the range of values that are all < 0.6 representatives of the region during no dust events or just the hours before the event as shown?**

Unfortunately, there are not enough studies or information to make a conclusion or generalize these values. We are unsure if this average is representative of the entire area as we only examined month-long data with a focus on one dust event. And all sensors were concentrated over several locations, and not spread enough, so we are unsure if they are representative of the entire area. While the monthly average does show lower values compared to 0.6, the low monthly values could likely be driven by the number of dust events that occurred during that month. We did however observe daily values above 0.6. mainly in sensors that were located in large urban areas. That information was added to the revised manuscript:

*Most of the stations had lower monthly values compared to the daily $PM_{2.5}/PM_{10}$ values, some stations had similar values of ~1.*

**Conclusion**
**It's very interesting that the PM2.5 fraction is so low during the event, much lower than dust events in other areas. This should be mentioned here as well as in the results.**

Per this reviewer's comments, we added more information to the revised manuscript that discusses this aspect. We found that the values were similar to previous studies, so we added that information as well as a comparison to other locations to the revised manuscript:

*Observations based on $PM_{2.5}/PM_{10}$ were also performed. The $PM_{2.5}/PM_{10}$ ratio is an important indicator used to characterize the underlying atmospheric processes within the local environment, which allows for the identification of the source of the particles (Yu and Wang, 2010). Higher $PM_{2.5}/PM_{10}$ ratios (> 0.6) are generally associated with anthropogenic pollution, while lower ratios are associated with dust events (Jugder et al., 2014; Sugimoto et al., 2016; Jaafari et al., 2018; Fan et al., 2021; Ardon-Dryer et al., 2022b). $PM_{2.5}/PM_{10}$ values across the nine sensors decreased during the dust event mainly between 11:00 to 18:00 LT (Fig. 6). $PM_{2.5}/PM_{10}$ values across the nine stations ranged from 0.03 to 0.13 with an average of 0.07 ± 0.02 across all stations and times. These ratios were lower compared to the values reported by Tong et al. (2012), which were 0.22-0.24, for this area combined with multiple dust events. Since Tong et al. (2012) $PM_{2.5}/PM_{10}$ values were based on daily values calculations of daily values for each sensor were made (Table S5). The daily $PM_{2.5}/PM_{10}$ values were in the same range (and even slightly higher, 0.24 -0.3) as those in Tong et al. (2012). However, observations of these ratios during the time of dust (which were shorter than the duration of the day, as discussed above) were lower, with the average $PM_{2.5}/PM_{10}$ value of 0.07 (values across all stations ranged from 0.05 to 0.09). These values were similar to those measured by Li et al. (2005) during dust events in the El Paso region. The hourly $PM_{2.5}/PM_{10}$ values at the peak of the dust were lower compared to those measured at the peak in multiple dust storms in Utah (Hahnenberger and Nicoll, 2012; Nicoll et al., 2020). In Washington state, a similar range of daily $PM_{2.5}/PM_{10}$ values was measured (Claiborn et al., 2000). The daily $PM_{2.5}/PM_{10}$ values were in a similar range to those measured during dust events around the world (Alghamdi et al., 2015; Malaguti et al., 2015; Sugimoto et al., 2016; Jaafari et al., 2018).*

**Is it known if this is a regional signature or is this not well-characterized since most sites don't have PM10 and PM2.5?**

Unfortunately, we cannot indicate this conclusion since there are only a handful of sensors that had both $PM_{10}$ and $PM_{2.5}$ measurements at the same site. And there have not been many measurements with both $PM_{10}$ and $PM_{2.5}$ measurements during dust in this region. We added more information to the revised manuscript per the reviewer's comments:

*It seems that in some of the locations, the contribution of coarse particles was more crucial than those of fine particles as shown by the low $PM_{2.5}$ and high $PM_{10}$ concentrations, and by the high $PM_{10}$-$PM_{2.5}$ values and low $PM_{2.5}/PM_{10}$ ratios (at least for the stations that had measurements for both $PM_{2.5}$ and $PM_{10}$). However, several of the stations showed higher $PM_{2.5}$ concentrations during the dust events, even 5 times higher (as C1028, in Lubbock). This location and many of the others only contain measurements of $PM_{2.5}$ leading to speculation if the lower contribution for $PM_{2.5}$ would be across the region or just in sites examined (the majority of them were in an urban site). Additional studies are needed during dust events and dust storms across the region to provide measurements for both $PM_{10}$ and $PM_{2.5}$. Additional measurements of particle size distribution are important, as such information will give critical knowledge related to health impact (inhalation of particles into the respiratory system), as well as on radiation and perhaps on cloud formation and precipitation processes.*

**What might the implications be to human health, atmospheric transport and potentially climate for dust events like these versus those that have a higher PM2.5/PM10 ratio – or – is there so much total mass that the PM2.5 fraction is not to be overlooked? How do the storms referenced in the results with ratios > 0.6 relate to this storm in terms of PM concentrations?**

We believe the request by the reviewer is beyond the scope of this study, as this study focuses on one single dust event. We believe what the reviewer is asking is a comparison between multiple dust events that will show different ratio values, which we could not provide for this study. According to the literature, dust events should not have a ratio <0.6, these rations will represent anthropogenic pollution, but as indicated additional studies are needed as there is not much information on the topic especially in this region.

This information was added to the revised manuscript to reflect this comment:

*This location and many of the others only contain measurements of $PM_{2.5}$ leading to speculation if the lower contribution for $PM_{2.5}$ would be across the region or just in sites examined (the majority of them were in an urban site). Additional studies are needed during dust events and dust storms across the region to provide measurements for both $PM_{10}$ and $PM_{2.5}$. Additional measurements of particle size distribution are important, as such information will give critical knowledge related to health impact (inhalation of particles into the respiratory system), as well as on radiation and perhaps on cloud formation and precipitation processes.*

**If a lot is not known about what meteorological conditions and seasons result in observed dust events and storms for this region, perhaps the conclusions should include a future work section that discusses what is known versus what needs more research, and whether this can**

**be done with the existing observations or what might improve data collections to further the state-of-the-science for the scientific community to understand dust events in the southwestern US as they are now and might change in the future.**

Per the reviewer's comment, we added information on the meteorological condition's aspect. Since we did not want to extend the conclusion section too long and beyond the scope of the work, we added one aspect related to the meteorological conditions. But in the main manuscript, we added more ideas for future work.

This information was added to the revised manuscript

*This location and many of the others only contain measurements of $PM_{2.5}$ leading to speculation if the lower contribution for $PM_{2.5}$ would be across the region or just in sites examined (the majority of them were in an urban site). Additional studies are needed during dust events and dust storms across the region to provide measurements for both $PM_{10}$ and $PM_{2.5}$. Additional measurements of particle size distribution are important, as such information will give critical knowledge related to health impact (inhalation of particles into the respiratory system), as well as on radiation and perhaps on cloud formation and precipitation processes.*

*Perhaps the meteorological conditions that initiated the dust for Lubbock (synoptic with convective) led to these high PM concentrations. Additional studies across the region are needed to understand how meteorological conditions that initiate dust events might impact the PM concentrations, as such information could be critical for prediction purposes which will help alert the public.*

**Specific comments:**
**Lines 151 and 334: Was the period "16 hours" or 18 that was stated in the abstract?**

We thank the reviewer for finding this mistake the numbers were corrected to 16

**Line 274: type-o - replace "tru exsposure" with "true exposure"**

We thank the reviewer for finding this mistake, correction was made.

**Tables S3 and S4 – State in the table header or footer what the bold data signifies.**

This information was added to the revised manuscript to reflect the reviewer's comment, in the title and also under the table, sown in previous comments (pictures of Tables)

These are the new titles of these tables
*Table S3. Meteorological parameters measured by ASOS stations during the dust event. Duration of the dust storm (DS) reported only for stations that reported visibility below 1 km. Bold numbers represent stations with visibilities < 1 km therefore DS.*

*Table S4. Measurements of $PM_{2.5}$ and $PM_{10}$ during the 26 February dust storm and the month of February 2023. Bold numbers represent significant $R^2$ values.*

---

## Author Comment (AC2)

Dear Editor,

Thank you for agreeing to consider a revision of our manuscript "The Spatial, Temporal, and Meteorological Impact of the 26 February 2023 Dust Storm, Increase of Particulate Matter Concentrations Across New Mexico and West Texas ". We modified and revised the manuscript to address the reviewers' comments as well as to clarify points that they found confusing or unclear.

We would like to thank the two reviewers, Dr. Allison C. Aiken and the anonymous reviewer for their helpful comments and suggestions, and many thanks to you for your time and efforts with this revision. In line with the comments and suggestions, we revised the manuscript and made the requested additions and changes. Below are all the comments (in bold) followed by the replies. The parts that are in italics are corrections that are included in the revised version of the paper:

Sincerely,
Karin Ardon-Dryer

**Review of "The Spatial and Temporal Impact of the Dust Storm During February 26, 2023, on Meteorological Conditions and Air Quality Across New Mexico and West Texas" by M.C. Robinson, K. Schueth, and K. Ardon-Dryer**
**General Comments: This paper examines the meteorological conditions and air quality impacts of a severe dust storm in New Mexico and West Texas in February 2023. Multiple observational datasets are used to characterize the features of the event and associated weather conditions. It is found that the upper-level jet streak, the passage of a cold front, and the formation of thunderstorms along the dryline all contribute to the high wind speeds during the event. The resultant visibility reduction and dramatic increase in PM values highlight the severity of the event. Overall, the study provides a timely and detailed analysis of an extreme dust storm (e.g., the highest PM2.5 record at Lubbock, Texas, in the past 20 years), which can potentially advance the current understanding of severe dust storms in the southern U.S. However, I found a few aspects that can be further improved. See my comments below for details.**

We would like to thank the reviewer for the suggestions, corrections, and comments.

**Specific comments:**
**1. The introduction section can be improved by adding a brief review of dust storms in the southwestern U.S., particularly over New Mexico and western Texas, as background information and by adding a few lines to highlight the motivation and novelty of this study. For instance, what are the key research questions that would be addressed in this study?**

Per the reviewer's comments, additional information was added to the introduction in order to provide more information on dust in the southwestern U.S., particularly over New Mexico and western Texas. The last paragraph of this introduction was modified to reflect the reviewer's comment.

These parts were added to the revised manuscript:

[revised manuscript text omitted]

**3. I think Figs. S1-S4 contain information that helps better understand the analysis and should be moved to the main text, given that currently only four figures are in the main text.**

These figures were moved from the supplements into the main manuscript. We decided to leave only one figure in the supplement as with decided its contribution was not crucial to the paper.

**4. It would be interesting to add some analysis or discussion about the physical mechanisms and unique aspects of this dust storm, for instance, what caused the strong winds? Lines 131-133 provide some discussion, but it would be interesting to show more if possible.**

We believe the original manuscript provided all the possible information on the physical mechanisms that describe why the dust event was initiated. The stacked jets and mixing of winds to the surface were the main physical meteorological reasoning as to why there were strong winds. This meteorological setup was not unique to this area, but it was rare to see these intense stacked lows and mixing of strong winds to the surface during the morning hours, which started to loft dust particles before the Pacific front which just intensified the dust.

Per the reviewer's comment, we added more information to the revised manuscript:
*The fact that some of the locations had both synoptic and convective disturbances (also known as combinations) is a rare aspect of this region, as only a handful of the dust events were caused by such conditions, for the case of Lubbock Texas, ~15% of the past DS (2000-2021) were caused by a combination of disturbances (Robinson and Ardon-Dryer, 2024).*

**5. Line 49, airports over certain regions, or the whole U.S.?**
Information was added to the revised manuscript, these are airports across the whole U.S.
*Automatic surface observation systems (ASOS) are meteorological stations located at most airports across the United States that provide meteorological measurements….*

**6. Lines 57-58, a severe storm with heavy precipitation can also reduce visibility and increase surface wind but without any dust storms.**

We agree with the reviewer that precipitation will reduce visibility. However, information on Precipitation is measured by the ASOS and provided in the METAR report, and it is easy to see it. We do not include such cases in our analysis as the reduction of visibility might be caused by the precipitation and not the dust particles. When precipitation occurs during dust events, the first observation of precipitation will be the end of the dust event. The precipitation will end up increasing the visibility as it clears the dust. Per the reviewer's comment, we added the aspect of precipitation to the sentence to clarify that we did not use that as part of our analysis. In this dust storm, none of the meteorological stations used in this study had precipitation during or after the dust.

This information was added to the revised manuscript:
*The classification of the dust event in this study was based on the combination of present weather codes such as BLDU (blowing dust), VCBLDU (vicinity blowing dust), DU (widespread dust), DS (dust storm), and HZ (haze), with the reduction of horizontal visibility (< 10 km) and increase of wind speed (> 6 m s$^{-1}$) but without precipitation, similar to the method used in Ardon-Dryer et al. (2023b) and Robinson and Ardon-Dryer (2024).*

**7. Section 2.3, why is the RAPv3 selected for the analysis? What variables are used?**

The RAPv3 was selected to illustrate meteorology due to its one-hour assimilation frequency and ability to provide one of the best forecasts in the rapidly changing atmosphere. Information on the variables used was also added to the revised manuscript.

*The synoptic maps were made using the North American Rapid Refresh version 3 (RAPv3) with a horizontal grid spacing of 13 km and 51 vertical levels (Benjamin et al., 2016). The RAPv3 was selected to illustrate the meteorology due to its one-hour assimilation frequency and ability to provide one of the best forecasts in the rapidly changing atmosphere. Only the initialization hours and no forecast hours were used in this study. Each synoptic map was made using the Metpy python package (May et al., 2023), with several meteorological variables layered. The following variables were chosen to analyze the meteorology; geopotential heights (mid-level and surface), wind speed and direction (mid-level and surface), temperatures (mid-level), and dewpoint temperatures (surface).*

**8. Lines 129-130, which figure do you refer to?**

Information on the figure was added to the revised manuscript

**9. Line 142, can you please provide definitions for 'blowing dust,' 'vicinity blowing dust,' and 'dust storm'?**

Information on these weather codes was added to the revised manuscript:
*The classification of the dust event in this study was based on the combination of present weather codes such as BLDU (blowing dust), VCBLDU (vicinity blowing dust), DU (widespread dust), DS (dust storm), and HZ (haze), with the reduction of horizontal visibility (< 10 km) and increase of wind speed (> 6 m s$^{-1}$) but without precipitation, similar to the method used in Ardon-Dryer et al. (2023b) and Robinson and Ardon-Dryer (2024). The different present weather codes for dust are defined by the World Meteorological Organization (WMO) and the Federal Aviation Administration (FAA). BLDU represents a case when the dust is present in the atmosphere and visibility drops below 11 km, DU indicates that dust is present and gives distant objects a tan or gray tinge, DS represents when dust drops the visibility to 1 km or less, and VCBLDU refers to that the dust is present within 8 to 16 km away from the station. Each of these codes can only be entered manually by a weather observer (WHO, 2019; FAA,2021). It should be noted that 16.1 km is the maximum visibility that should be reported by the ASOS (ASOS User's Guide, 1998). Many studies have used the present weather codes to identify dust events in this region (Kandakji et al., 2020; Herrera-Molina et al., 2021; Kelley and Ardon-Dryer, 2021; Robinson and Ardon-Dryer, 2024).*

**10. Line 216, is there an upper limit of PM10 measurement?**

Yes, there is an upper limit to the instrument, and each might have a different one. Some of the PM$_{2.5}$ units had an upper limit of 5,000 µg m$^{-3}$ or 10,000 µg m$^{-3}$, and many of the PM10 units had an upper limit of 10,000 µg m$^{-3}$. it should be noted that we did not have control over these as each start air quality agency decided which unit to use and where. Information on the matter was added to the revised manuscript:

*All of the PM sensors are Federal Equivalent Methods (FEMs). Each FEM instrument had a different resolution depending on the operated unit (See Table S2), some units ranged from 0.1 up to 10,0000 μg m⁻³ (T640, 2024), or -15 up to 10,0000 μg m⁻³ (BAM 1022, 2024), other had an upper limit of 5,0000 μg m⁻³ (R & P Model 2025; EPA, 2024).*

**11. Line 265, is it the daily average of 26 Feb. 2023?**

We added information in the revised manuscript and in the supplement to clarify that the daily average refers to Feb 26[th]. Per comments from reviewer 2, we performed daily calculations every day from February 2023 for each PM sensor.

**12. Line 280, how are correlations calculated? Do you use hourly data of the day? Are autocorrelations considered?**

The regression used were linear and polynomials, both were based on hourly values of wind and PM from the stations used in this study. No autocorrelations were used in this study. Information was added to the revised manuscript and to Table S4.

*Calculations of regression (linear and polynomial) were made based on hourly PM concentrations and wind speeds for all stations with measurements from February 26.*

**13. Line 285, are the correlations significant?**

Some were significant but others were not, this information was provided in detail in the manuscripts as well as in Table S4 for each of the stations.

**14. The current title indicates that meteorological conditions are affected by the dust storm. However, the text suggests the other way around. Please consider rewording to avoid confusion.**

The title of the manuscript was modified per the comments from both reviewers, this is the title of the revised manuscript:
*The Spatial, Temporal, and Meteorological Impact of the 26 February 2023 Dust Storm, Increase of Particulate Matter Concentrations Across New Mexico and West Texas*

**15. In Figure 2, please consider reducing the density of contours. In Fig. 2a-b, is the shading total wind speed? Also, can you please add labels for temperature and geopotential height? Similarly, for Fig. 2c, please add labels for surface height. And why are these time snapshots, i.e., 18UTC on 26 Feb. and 00UTC on 27 Feb., selected? When did the storm start?**

Changes were made to the figure as recommended by the reviewer. The density of contours has

been reduced. The wind speed is shaded in Fig 2a-b and dewpoints are shaded in Fig 2c-d, which is now indicated in the caption. The time snapshots of 18UTC on 26 Feb. and 00UTC on 27 Feb. were chosen to show the evolution of the system the weather system that amplified the dust storm. The two times chosen are considered synoptic times and best represent the evolution through the afternoon and early evening hours, which is when the dust storm intensified across West Texas.

These changes were made in the revised manuscript:

[Figure]

*Figure 2. 500 mb geopotential heights (m), wind speed (kt, shaded), wind barbs (kt), and temperature (°C) for February 26 at 18:00 UTC, 12:00 central time, when the dust started (A) and 27 at 00:00 UTC, 18:00 central time, when the dust intensified across west Texas (B) and surface wind barbs (mph) and dew point temperature (°C, shaded) for February 26 at 18:00 UTC, 12:00 central time (C) and 27 at 00:00 UTC, 18:00 central time (D).*

---

## Author Response (AR2)

Dear Editor Prof Barbara Ervens

Thank you for agreeing to handle our manuscript and for considering a revision of our manuscript. We modified and revised the manuscript to address the editor's comments. We revised the manuscript and made the requested changes in line with your suggestions. Below are all the comments (in bold) followed by the replies. The parts that are in italics are corrections that are included in the revised version of the paper:

Sincerely,
Karin Ardon-Dryer

**Public justification (visible to the public if the article is accepted and published**): **Dear Authors,**

**Thank you for revising your manuscript. Both referees are satisfied with the changes. However, I have a few minor comments (see below) that I would like you to address carefully. Once these are resolved, I will be happy to accept your paper for publication in ACP. In case you are wondering about the change in editor during the review process: the previous editor decided to step down from the editorial board. As one of the executive editors, I took over to avoid any further delays.**

**Best regards**
**Barbara Ervens**

**=========================**
**- The title could be further improved, e.g. "Spatial, Temporal, and Meteorological Impacts of the February 26, 2023 Dust Storm: Increase in Particulate Matter Concentrations Across New Mexico and West Texas"**
The tile was modified as suggested.

**- Please check the ACP author guidelines https://www.atmospheric-chemistry-and-physics.net/policies/guidelines_for_authors.html**
**1) If possible, please shorten the abstract by about 10 words**
We are a bit confused according to ACP guidelines "Abstracts should have fewer than 250 words" and our abstract had 248 words

**2) Pay attention to the guidelines for the concluding section. It should exceed a summary of the results.**
The conclusions section was modified per the editor's comments.

*On 26 February 2023, an upper-level low-pressure system with a strong jet streak aided in the mixing of strong winds to the surface, which resulted in the formation of a dust storm over portions of New Mexico and West Texas. The dust first initiated in New Mexico during the morning hours and intensified as it moved eastward into West Texas. The average wind speed at the beginning of the dust storm was $15.6\ m\ s^{-1}$ and during the dust storm wind speeds reached up to $26.2\ m\ s^{-1}$ with wind gusts up to $37\ m\ s^{-1}$. Similar wind speeds were measured during different dust storms across the Great Plains, yet lower wind speeds were measured during several dust storms in Arizona. Visibilities ranged from 4 km down to 0 km defining the event as a dust storm (visibility < 1 km). 11 ASOS stations reported dust storm conditions for about 5 to 120 minutes, and Lubbock ASOS reported zero visibility for 13 minutes. This dust storm had a big impact on the air quality in the area. Daily PM concentrations that exceeded the EPA daily threshold ranged from $36 \pm 40\ \mu g\ m^{-3}$ up to $69 \pm 121\ \mu g\ m^{-3}$ for $PM_{2.5}$ and $205 \pm 321\ \mu g\ m^{-3}$ up to $748 \pm 2090\ \mu g\ m^{-3}$ for $PM_{10}$. Nine PM stations exceeded the EPA daily threshold. High hourly $PM_{2.5}$ and $PM_{10}$ concentrations during the dust storm reached a maximum of $518\ \mu g\ m^{-3}$ and $9983\ \mu g\ m^{-3}$ respectively. $PM_{10}$-$PM_{2.5}$ at the time of the dust, based on nine stations ranged from $96 \pm 61\ \mu g\ m^{-3}$ up to $760 \pm 1000\ \mu g\ m^{-3}$, which is approximately 6 times higher than the daily $PM_{10}$-$PM_{2.5}$ values and 12 times higher than monthly $PM_{10}$-$PM_{2.5}$ values. $PM_{2.5}/PM_{10}$ during the dust time, ranged from $0.05 \pm 0.01$ up to $0.09 \pm 0.03$, which were 3.6 times lower than the daily and monthly $PM_{2.5}/PM_{10}$ values. The PM stations in the region, especially in West Texas, are spaced and far apart meaning that higher PM concentrations than those measured could have occurred but not been reported. Dust particles were present in the air for approximately 16 hours impacting millions of citizens across eastern New Mexico and West Texas. In some locations (e.g., Lubbock), this dust storm was the strongest ever reported, as it had the highest $PM_{2.5}$ concentrations recorded since the station became operational in 2001 and the lowest visibility recorded during a dust storm since 2003. Perhaps the meteorological disturbances that initiated the dust for Lubbock (synoptic with convective) led to these high PM concentrations. Additional studies across the region are needed to understand how meteorological disturbances that initiate dust events might impact the PM concentrations, as such information could be critical for prediction purposes which will help alert the public. Such information could determine whether long-term effects such as land usage and climate change will affect the frequency and intensity of dust storms in this region.*

**l. 14: Swap '< 10 and 2.5 µm' to '< 2.5 and 10 µm' to be consistent with order in preceding 'PM2.5 and/or PM10'**

The changes were made as suggested.

**l. 27: What is meant here by 'dust initiation'? DO you mean 'initiation of dust events and storms'?**

The sentence was changed as we also added additional references to support it.

*Strong winds are very important for the initiation of dust events and/or storms, which are generally caused by a synoptic or convective meteorological disturbance (Kelley and Ardon-Dryer, 2021; Robinson and Ardon-Dryer et al., 2024; Sandhu et al., 2024).*

**l. 64: 68.5 ± 72 should be rounded to 69 ± 72. Make sure to use the same number of significant digits also at other places in the manuscript.**

The changes were made as suggested throughout the manuscript.

**l. 85: How do you define 'significant' in this context? If it doesn't add quantitative and/or statistically relevant information here, I suggest removing the word.**

The words "and significance" were removed from the sentence as suggested by the editor.

**l. 126: Where in Table S2 can the reader see the resolution of the instruments?**

The sentence was modified to reflect this comment.

*Each FEM instrument had a different resolution depending on the operated unit, some units ranged from 0.1 up to 10,0000 µg m⁻³ (T640, 2024), or -15 up to 10,0000 µg m⁻³ (BAM 1022, 2024), others had an upper limit of 5,0000 µg m⁻³ (R & P Model 2025; EPA, 2024), see Table 2 for information on instrument used at each location.*

**l. 127/128: Is there a '0' too much in 10,0000 and 5,0000, or do you indeed mean one hundred thousand and fifty thousand?**

We thank the editor for catching this error we modified the numbers to reflect the correct ones.

**l. 152/3: "Only the initialization hours and no forecast were hours used in this study." This is unclear. Please clarify this sentence.**

The sentence was modified to reflect this comment

*Only the initialization hours were hours used in this study.*

**l. 174: To me, it seems unusual to call 18:00 'early to mid afternoon' – or is it a different time zone you are referring to? Please clarify.**

We thank the editor for finding this mistake, the hours were removed from the sentence.

*The right exit region (Fig. 2B) of the nearly 51-62 m s⁻¹ (100-120 knot) 500 mb jet streak, associated with the upper low, entered the Chihuahuan Desert region of Mexico, Texas, and New Mexico around early to mid-afternoon.*

**l. 205 (Fig 2 caption): 'when the dust started', 'when the dust intensified across west Texas' – please rephrase using 'dust event' or 'dust storm'.**

The caption was modified to reflect the editor's comment.

*Figure 2. 500 mb geopotential heights (m), wind speed (kt, shaded), wind barbs (kt), and temperature (°C) for 26 February 2023 at 18:00 UTC, 12:00 central time, when the dust event started (A) and 27 at 00:00 UTC, 18:00 central time, when the dust event intensified and turned into a dust storm across west Texas (B) and surface wind barbs (mph) and dew point temperature (°C, shaded) for February 26 at 18:00 UTC, 12:00 central time (C) and 27 at 00:00 UTC, 18:00 central time (D).*

**l. 228: 'started' might fit better here than 'initiated'**

The changes were made as suggested.

**l. 238/9: 5:36 ± 3:31 hours – can this be indeed said with this accuracy?**
These numbers were calculated based on the observation times, per the editor's comment the number was modified.
*On average the dust across all stations lasted for 5:30 ± 3:30 hours since some.......*

**l. 336: "The duration when dust particles were in the air based on an increase in PM values was similar to the duration based on visibility". This sentence is not fully clear. What do you mean by 'duration'? Is it a fixed, quantitative term that can be derived based on PM concentrations and visibility?**
The sentence was modified to reflect the editor's comment
*The duration of dust particles in the air was based on the time from the first increase in PM to the decrease in PM values. This duration was similar to the duration of reduction of visibility, mentioned in section 3.2.*

**l. 375 ff: What is the reasoning for a polynomial fit? Why would you expect a better (?) correlation than just by assuming a linear relationship between PM and wind speed? Some explanation for choice of this fit would be appreciated.**
We believe this information is provided later in the paragraph. While the normal aspect will be to look at linear regression because the increase in some stations was so high we examined also other regression tests to try to find which can help explain the changes in wind speed and PM value.

This information is provided in lines 384-388: *Other regression models were also examined, to potentially find a better regression value between wind speed and PM values. The Polynomial regression (with 2nd-degree polynomial) presented much higher $R^2$ values compared to a linear regression. With $R^2$ values that ranged from 0.37 up to 0.9 for $PM_{2.5}$ and from 0.18 up to 0.9 for $PM_{10}$. 73.3% of the $PM_{2.5}$ stations and 84.6% of the $PM_{10}$ stations had $R^2 \geq 0.5$ (see $R^2$ values in Table S4).*

**l. 442/3: "These high hourly concentrations are imported as studies,…" – What do you mean by 'imported'? Please clarify.**
The sentence was modified to reflect the editor's comment
*It is important to have hourly concentration measurements, as studies during dust events from this region (El Paso and Lubbock) have shown that maximum daily PM concentrations can lead to significant increases in hospitalizations on the day of dust and the following days.*

**l. 451: I assume you mean the difference between PM2.5 and PM10 by (PM10-PM2.5). It might be easier to read in the subsequent text if you defined a parameter for it, e.g. dPM or deltaPM = PM10 – PM2.5**

We thank the editor for the suggestion, but we do not want to confuse the reader, based on the literature and papers that use this ratio the term PM10-PM2.5 is commonly used. We do not want to start using a different term that is not commonly used which might raise questions and concerns for the readers.

**Supplement: Please add the table captions to the tables.**

Table captions were added to the Excel file in each tab for each of the tables.